

# Population genetic structure of Patagonian toothfish (*Dissostichus eleginoides*) in the Southeast Pacific and Southwest Atlantic Ocean

Cristian B. Canales-Aguirre[1,2,3,4], Sandra Ferrada-Fuentes[2,5], Ricardo Galleguillos[2], Fernanda X. Oyarzun[1,6] and Cristián E. Hernández[3]

[1] Centro i~mar, Universidad de Los Lagos, Camino Chinquihue Km 6, Puerto Montt, Chile
[2] Laboratorio de Genética y Acuicultura, Departamento de Oceanografía, Universidad de Concepción, Concepción, Chile
[3] Laboratorio de Ecología Evolutiva y Filoinformática, Departamento de Zoología, Universidad de Concepción, Concepción, Chile
[4] Núcleo Milenio INVASAL, Concepción, Chile
[5] Programa de Doctorado en Sistemática y Biodiversidad, Universidad de Concepción, Concepción, Chile
[6] Centro de Investigación en Biodiversidad y Ambientes Sustentables (CIBAS), Facultad de Ciencias, Universidad Católica de la Santísima Concepción, Concepción, Chile

Corresponding author
Sandra Ferrada-Fuentes,
sferrada@udec.cl

## ABSTRACT

Previous studies of population genetic structure in *Dissostichus eleginoides* have shown that oceanographic and geographic discontinuities drive in this species population differentiation. Studies have focused on the genetics of *D. eleginoides* in the Southern Ocean; however, there is little knowledge of their genetic variation along the South American continental shelf. In this study, we used a panel of six microsatellites to test whether *D. eleginoides* shows population genetic structuring in this region. We hypothesized that this species would show zero or very limited genetic structuring due to the habitat continuity along the South American shelf from Peru in the Pacific Ocean to the Falkland Islands in the Atlantic Ocean. We used Bayesian and traditional analyses to evaluate population genetic structure, and we estimated the number of putative migrants and effective population size. Consistent with our predictions, our results showed no significant genetic structuring among populations of the South American continental shelf but supported two significant and well-defined genetic clusters of *D. eleginoides* between regions (South American continental shelf and South Georgia clusters). Genetic connectivity between these two clusters was 11.3% of putative migrants from the South American cluster to the South Georgia Island and 0.7% in the opposite direction. Effective population size was higher in locations from the South American continental shelf as compared with the South Georgia Island. Overall, our results support that the continuity of the deep-sea habitat along the continental shelf and the biological features of the study species are plausible drivers of intraspecific population genetic structuring across the distribution of *D. eleginoides* on the South American continental shelf.

## INTRODUCTION

The long-held idea that the deep-sea environment is composed of spatially homogeneous habitats that remain stable for long periods of time (*Gooch & Schopf, 1973*) led to the assumption that populations of deep-sea animals had low genetic variability. Consequently, it has often been assumed that speciation in the deep-sea occurred as a result of geographic isolation-by-distance (IBD; *Wilson & Hessler, 1987*). The apparent homogeneity of the marine environment (e.g., *Bunawan et al., 2015*; *Magallón-Gayón, Diaz-Jaimes & Uribe-Alcocer, 2016*) and the many dispersal mechanisms of marine organisms has led to the idea that most marine populations are open populations (*Cowen et al., 2000*; *Hedgecock, Barber & Edmands, 2007*; *Cowen & Sponaugle, 2009*). Microevolutionary studies in deep-sea organisms have revealed that geographical gradients and bathymetry play an important role in population genetic structure (*Zardus et al., 2006*; *Jennings, Etter & Ficarra, 2013*; *Porobić et al., 2013*; *Baco et al., 2016*; *Shen et al., 2016*).

In broadly distributed benthopelagic fishes, considerable gene flow has been reported among populations. Scarce genetic divergence is therefore mainly the result of the availability and continuity of their habitats (e.g., slopes of continents slopes, oceanic islands, and seamounts), facilitating gene flow (*Smith & Gaffney, 2005*; *Jones et al., 2008*; *Lévy-Hartmann et al., 2011*; *Varela, Ritchie & Smith, 2012*). In addition, biological features such as vagile and/or pelagic adults and long-duration planktonic eggs, larvae and/or juvenile stages are associated with low intraspecific genetic differentiation (*Shaw, Arkhipkin & Al-Khairulla, 2004*; *Rogers et al., 2006*). For example, gene flow has been reported in *Chaenocephalus aceratus*, *Notothenia coriiceps*, and *Lepidonotothen larseni* distributed in the Southern Ocean (*Jones et al., 2008*), as well as in *Dissostichus mawsoni* (*Smith & Gaffney, 2005*), and even in cosmopolitan species from seamounts such as *Hoplostethus atlanticus* (*Varela, Ritchie & Smith, 2012*), and *Beryx splendens* (*Lévy-Hartmann et al., 2011*).

The Patagonian toothfish, *Dissostichus eleginoides* Smitt, 1898, is the most productive and lucrative fishery in the entire Antarctic, Southern Ocean, and southern portions of the oceans around the southern South American cone. This species is vulnerable to overfishing because of its size, long life span, relatively small numbers of eggs and delayed onset of reproductive maturity (*Bialek, 2003*). *Dissostichus eleginoides* can reach 2 m in length, becomes sexually mature around 7–12 years, can live up to 30 years (*Laptikhovsky, Arkhipkin & Brickle, 2006*), and has low fecundity in relation to its body weight (*Young, Gill & Cid, 1995*). The Patagonian toothfish is distributed in cooler waters between 70 and 2,500 m deep, although it is typically fished below depths of 200 m (*Evseenko, Kock & Nevinsky, 1995*). The genus *Dissostichus* belongs to the family Nototheniidae, a diverse clade of Antarctic and sub-Antarctic origin (*Bargelloni et al., 2000*; *Di Prisco et al., 2007*). *Dissostichus* has only two species, *D. mawsoni* and *D. eleginoides,* which diverged in the Miocene, 14.5 million years ago (*Near, 2004*). *Dissostichus eleginoides* has a discontinuous distribution restricted to seamounts and submarine platforms in sub-Antarctic waters, but a wide continuous distribution in the Southeastern Pacific continental shelf and slope (*Oyarzún & Campos, 1987*). The continuous distribution of this species along the South American continental shelf in the Southeastern Pacific Ocean could facilitate gene flow homogenizing their

population genetic structure, especially taking into account that *D. eleginoides* has pelagic early stages (*North, 2002*) and trophic-reproductive migrations throughout this area (*Laptikhovsky & Brickle, 2005*; *Laptikhovsky, Arkhipkin & Brickle, 2006*).

Population genetics studies of *D. eleginoides* to date have been mainly conducted in the Southern Ocean. Using allozyme and microsatellite loci, *Smith & McVeagh (2000)* showed that *D. eleginoides* has restricted gene flow between the Falkland Islands, and zones south of the Antarctic Polar Front (i.e., Heard Island, Ross Dependency, Prince Edward Island and Macquarie Island). Later, *Shaw, Arkhipkin & Al-Khairulla (2004)* showed that populations to the north of Antarctic Polar Front (i.e., Patagonian Shelf, North Scotia Ridge) and to the South of Antarctic Polar Front (i.e., Shag Rocks, South Georgia) have stronger genetic differentiation in mtDNA genome than the nuclear genome, based on microsatellites and mtDNA sequences. In a study conducted in the West Indian Ocean sector of the Southern Ocean, *Appleyard, Williams & Ward (2004)* investigated mtDNA and microsatellite loci but found no evidence for among-population genetic differences associated with islands. Subsequently, *Rogers et al. (2006)*, surveying samples from islands in the Atlantic, Pacific, and Indian Oceans, found genetic differences based on microsatellites and mtDNA data. Specifically, *Rogers et al. (2006)* indicated that toothfish populations from around the Falkland Islands were genetically distinct to those from around the South Georgia Island. Recently, *Toomey et al. (2016)* studied DNA from otoliths and found differences between populations around the Macquarie Island and others locations surveyed in the Southern Ocean.

All previous studies discussed above have focused mainly on islands of the Southern Ocean, leaving a distinct gap in our knowledge of the genetic structure of the *D. eleginoides* populations across their Southeastern Pacific Ocean distribution. The only study carried out in the Southeastern Pacific Ocean was developed by *Oyarzún et al. (2003)* based on allozymes and was restricted to a small geographic area. *Oyarzún et al. (2003)* did not find population genetic structure among samples collected in south-central Chile (c. 37°S to 43°S). Sampling across a wider geographical area of this region while using more sensitive molecular tools that have higher levels of detection of DNA polymorphism, such as microsatellite loci, could aid in determining whether or not significant population genetic structure exists among *D. eleginoides* populations across their Southeastern Pacific Ocean distribution.

In this study, we used a panel of six microsatellites previously developed for *D. eleginoides* to test whether this species shows population genetic structure on the South American Plateau. We hypothesized that *D. eleginoides* would show limited genetic structure due to the continuity of suitable habitats along the South American continental shelf, from Peru in the Pacific Ocean southward and eastward to the Falkland Islands in the Atlantic Ocean (Fig. 1).

## MATERIALS AND METHODS

### Ethics statement

*Dissostichus eleginoides* has not yet been assessed for the IUCN Red List and is not listed under CITES. Samples used in this study were collected in accordance with national
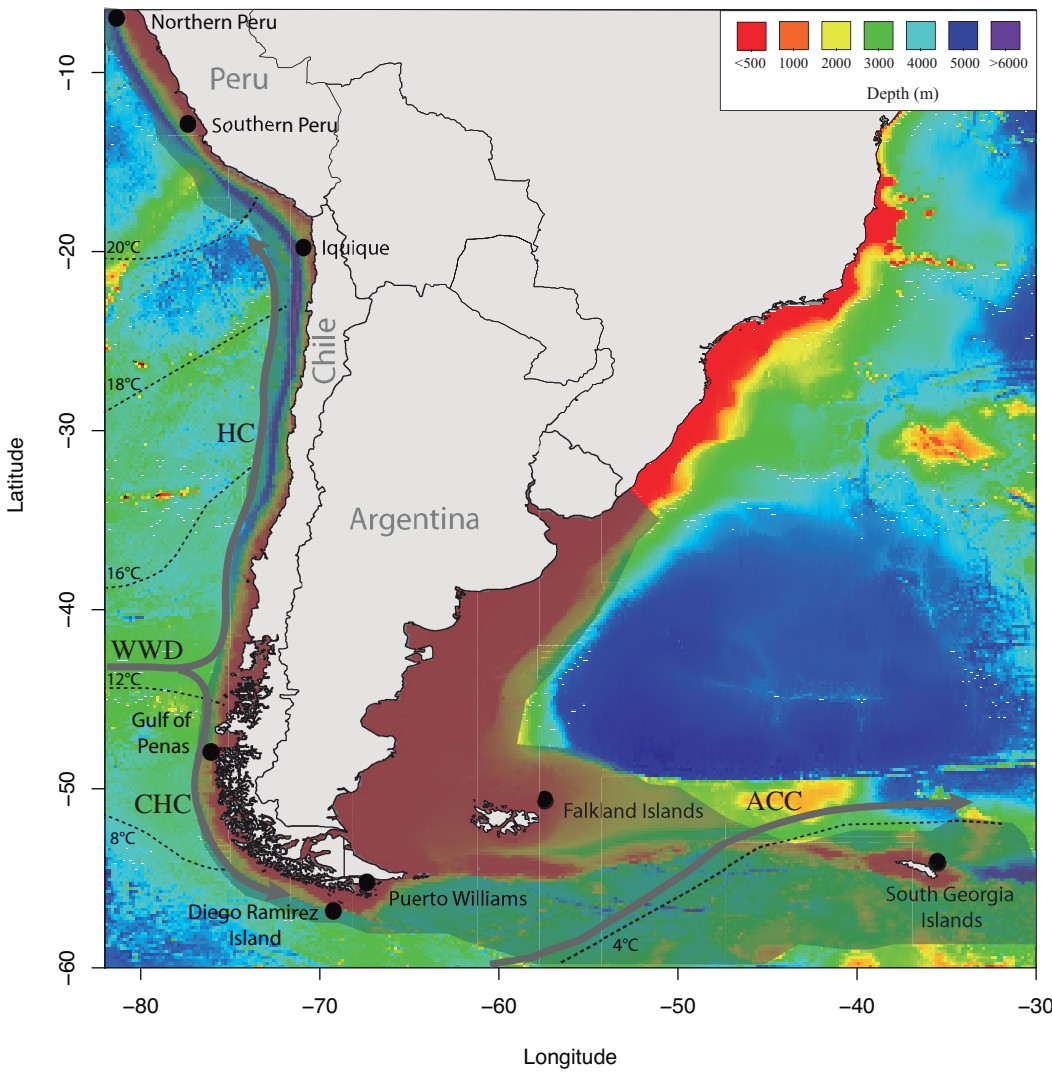

**Figure 1** **Map of sampling locations used in the present study showing the discontinuities in the southeastern Pacific and southwestern Atlantic Oceans.** Temperatures, currents, bathymetry of continental margin and deep-sea floor, and the distributional range of *D. eleginoides* on the South America continental shelf. The geographical distribution of *D. eleginoides* on the South American continental shelf was obtained from *Aramayo (2016)* and is shown in transparent gray shading. Dashed lines indicate mean annual sea surface temperatures redrawn from *Kaiser, Lamy & Hebbeln (2005)*. Bathymetries were obtained using the marmap R package (*Pante & Simon-Bouhet, 2013*). HC, Humboldt Current; WWD, West Wind Drift; CHC, Cape Horn Current; ACC, Antarctic Circumpolar Current.

legislation of the corresponding nations. In fact, no governmental approval of this vertebrate work was required since the Patagonian toothfish individuals sampled in this study were obtained from scientific and commercial fishing activities. We did not kill fishes for the purpose of this study; instead, we obtained tissue samples from individuals that were fished by authorized commercial vessels using long lines. Tissue samples of Patagonian toothfish used in this study were obtained from the Peruvian exclusive economic zone (EEZ) in collaboration with the Instituto del Mar del Peru (IMARPE). Tissue samples

**Table 1** Mean summary statistics for genetic variability, percentage of putative migrants, and effective population size by location and cluster inferred for *Dissostichus eleginoides*.

| Locality | Lat | Long | $N$ | $N_a$ | $H_O$ | $H_E$ | $M$ | $N_e{}^a$ | $N_e{}^b$ | $LDN_e$ | CI |
|---|---|---|---|---|---|---|---|---|---|---|---|
| NP | 7°35′ | 81°15′ | 27 | 15.0 | 0.781 | 0.779 | 3.7 | 244 | 24,421 | 182 | 69-Inf |
| SP | 12°46′ | 77°27′ | 25 | 14.5 | 0.738 | 0.750 | 8.0 | 188 | 18,803 | Inf | 328.3-Inf |
| IQ | 20°16′ | 70°49′ | 42 | 15.2 | 0.706 | 0.741 | 4.8 | 173 | 17,336 | 773 | 165-Inf |
| GP | 48°52′ | 75°25′ | 24 | 11.5 | 0.675 | 0.691 | 4.2 | 118 | 11,842 | 133 | 42.4-Inf |
| PW | 54°92′ | 67°62′ | 54 | 14.0 | 0.681 | 0.704 | 1.9 | 130 | 13,033 | 436 | 141.0-Inf |
| DRI | 56°30′ | 68°37′ | 66 | 15.3 | 0.709 | 0.720 | 3.0 | 147 | 14,694 | Inf | 375-Inf |
| FI | 49°34′ | 54°28′ | 48 | 16.5 | 0.754 | 0.740 | 4.2 | 172 | 17,241 | Inf | Inf-Inf |
| SGI | 54°39′ | 34°00′ | 71 | 12.7 | 0.758 | 0.650 | 2.8 | 90 | 8,954 | 188 | 99.5–852.6 |
| SAC | | NA | 286 | 22 | 0.720 | 0.745 | 0.7 | | 17,913 | 6,911 | 1,262.2-Inf |
| SGI | | NA | 71 | 13 | 0.758 | 0.650 | 11.3 | | 8,954 | 188 | 99.5–852.6 |

**Notes.**

Locality abbreviations: NP, Northern Peru; SP, Southern Peru; IQ, Iquique; GP, Gulf of Penas; PW, Puerto Williams; DRI, Diego Ramírez Islands; FI, Falkland Islands; SGI, South Georgia Island. Locality code SAC refers to the cluster including all locations that are on the South American continental shelf. The SGI cluster included individuals from the South Georgia Island.

Other abbreviations: Lat, Latitude; Long, Longitude; $N$, Number of individuals sampled; $N_a$, average of the number of alleles per locus; $H_O$, average of the observed heterozygosity; $H_E$, average of the expected heterozygosity; NA, not applicable; CI, confidence interval; Inf, infinite. The migrants ($M$) column shows the percentage of putative migrants from the first generation. Effective population size ($N_e$) was based on Linkage Disequilibrium ($LD$) (*Waples & Do, 2010*) and the *Nei (1987)* formula.

[a] Estimated using a mutation rate of $1 \times 10^{-2}$ (refs. in *DeWoody & Avise, 2000*).

[b] Estimated using a mutation rate of $1 \times 10^{-4}$ (refs. in *DeWoody & Avise, 2000*).

from Chilean EEZ were obtained during scientific research programs with the permission of the Chilean Fishery Government and obtained by the Instituto de Fomento Pesquero (IFOP). Additional tissue samples from the Falkland Islands and South Georgia Island were obtained from commercial long-liner vessels within the Total Allowed Catch quotas assigned by the Commission for the Conservation of Antarctic Marine Living Resources (CCAMLR).

## Sampling locations

A total of 417 individuals of *Dissostichus eleginoides* were sampled from a portion of the species range around South America and the South Georgia Island (Fig. 1), including the following locations (Table 1): NP, Northern Peru; SP, Southern Peru; IQ, Iquique; GP, Gulf of Penas; PW, Puerto Williams; DRI, Diego Ramírez Islands; FI, Falkland Islands; and SGI, South Georgia Island.

## Molecular and pre-processing genetics dataset

Total genomic DNA was isolated from samples of muscle tissue as described in *Grijalva-Chon et al. (1994)*. The quality and quantity of the isolated DNA was determined with an Eppendorf® BioPhotometer. Each sample was diluted in ultra-pure water at 20 ng /µl for PCR amplifications. Six microsatellite loci of *D. eleginoides* were used, cmrDe2, cmrDe4, cmrDe9, cmrDe30 (*Reilly & Ward, 1999*), To2, and To5; these loci have proven to be useful in differentiating populations of the Patagonian toothfish (*Smith & McVeagh, 2000*; *Appleyard, Ward & Williams, 2002*; *Shaw, Arkhipkin & Al-Khairulla, 2004*; *Rogers et al., 2006*). Microsatellite loci were amplified following the conditions described in *Appleyard, Ward & Williams (2002)*. PCR products were analyzed on an ABI 3730 automated

sequencer. Allele size was estimated in PEAKSCANNER$^{TM}$ v1.0 software with a GS500 internal weight marker.

We filtered out individuals that had more than two missing genotype loci, in order to avoid spurious results in the estimation of further genetic differentiation parameters (*Putman & Carbone, 2014*). Ultimately, we obtained a total data set of 357 individuals that we used in subsequent analyses. Afterwards, we estimated the presence of genotyping errors such as drop-out alleles, stutter bands, and likely presence of null alleles to evaluate the quality of the genetic database using the MICRO-CHECKER v2.2.3 software (*Van Oosterhout et al., 2004*). According to MICRO-CHECKER only 7 out of 48 tests may have exhibited null alleles due to an excess of homozygotes (Table S1), and none of the other genotyping errors were observed. Based on the algorithms described by *Brookfield (1996)* only one locus in the location GP (i.e., De2) showed an estimated null allele frequency over 10% (Table S1). *Chapuis & Estoup (2007)* proposed that null frequencies below 5% have a negligible impact on genetic differentiation analyses, however we performed further analyses with model-based clustering and Bayesian assignment methods (*Guillot, Santos & Estoup, 2008*; *Carlsson, 2008*) which take into account null alleles and significantly improve estimation accuracy (i.e., GENELAND, *Guillot, Santos & Estoup, 2008*). Finally, in order to avoid inflating patterns of genetic structure due to kinship control (i.e., effect of sampling families), we ruled out putative total kinship within samples for each location. To estimate total kinship, we use the maximum-likelihood method implemented in COLONY v2.0.0.1 (*Wang, 2004*; *Jones & Wang, 2010*). Total kingship analysis was conducted using the 'long length of run' and 'high likelihood precision' options implemented in COLONY. Results from the total kinship identification analysis did not show evidence for putative total kinship in the data set; therefore, we proceeded with data analyses without excluding any individuals. Raw data (i.e., multilocus genotypes) used for further analyses are included as Data S1.

## Genetic variability and population structure

We estimated the number of alleles ($N_a$), expected ($H_E$), and observed ($H_O$) heterozygosity to determine the genetic variability of the population surveyed; these population summary statistics were calculated for each locus and population using GENALEX v6.5 software (*Peakall & Smouse, 2012*). We tested significant deviation from Hardy-Weinberg equilibrium (HWE) by testing the hypothesis that the observed diploid genotypes are product of a random union of gametes using ARLEQUIN v3.5 (*Excoffier & Lischer, 2010*). This procedure was carried out locus-by-locus using the following parameter settings: 100,000 steps in the Markov chain and 10,000 dememorizations. In addition, we tested linkage disequilibrium (LD) association by testing the hypothesis that genotypes at one locus are independent from genotypes at another locus using GENEPOP v3.1 (*Raymond & Rousset, 1995*; *Rousset, 2008*). The parameters used in the Markov chain were: 1,000 dememorizations, 100 batches, and 1,000 iterations per batch. No pair of loci in our data set exhibited significant LD, which indicated that all the loci used in this study were independent one another (unlinked). We obtained $F_{ST}$ and $R_{ST}$ pairwise indices in ARLEQUIN to estimate the degree of genetic differentiation among samples locations.

The probability values for $F_{ST}$ and $R_{ST}$ were obtained by permutation tests with 10,000 replicates. We applied the sequential Bonferroni correction for multiple comparisons (*Rice, 1989*) when necessary.

## Number of clusters and isolation-by-distance

To infer the most likely number of genetic clusters ($K$) present in our data set, we used two Bayesian clustering methods, one in the program GENELAND v1.0.7 (*Guillot et al., 2005*; *Guillot, Mortier & Estoup, 2005*; *Guillot, Santos & Estoup, 2008*) and the other implemented in STRUCTURE v2.3.4 (*Pritchard, Stephens & Donnelly, 2000*; *Falush, Stephens & Pritchard, 2003*). GENELAND uses a Bayesian statistical population algorithm to model a set of georeferenced individuals with genetic data, while accounting for the presence of null alleles in the sample. The number of clusters was determined by 10 independent Markov chain Monte Carlo (MCMC) searches, which allowed us to estimate $K$ using the following parameters: $K$ from 1 to 8 (which is equivalent to the number of sampling locations surveyed in this study), $5 \times 10^6$ MCMC iterations, a thinning interval of 1,000, the maximum rate of process Poisson fixed at 357, and the maximum number of nuclei in the Poisson-Voronoi tessellation fixed at 1,071. Following recommendations of *Guillot, Santos & Estoup (2008)*, we ran the analyses using the uncorrelated frequency allele model because of the unknown number of $K$ in the study area, the spatial model, and the null allele model. Finally, we plotted a map of South America over the output of GENELAND, in order to visualize the results in the context of geography.

Although STRUCTURE does not include a null allele model and uses a non-spatial model based on a clustering method, it is useful for quantifying the proportion of each individual genome from each inferred population in $K$. The number of clusters was determined by performing ten runs with 50,000 iterations, followed by a burn-in period of 5,000 iterations, for $K = 1$–9. All STRUCTURE runs were carried out with an admixture model of ancestry, an independent allele frequency model, and a LOCPRIOR model (*Hubisz et al., 2009*). We incorporated Evanno's index $\Delta K$ (*Evanno, Regnaut & Goudet, 2005*) in order to identify the best $K$ value for our data set, using STRUCTURE HARVESTER (*Earl & VonHoldt, 2012*). Then, we plotted 'consensus' coefficients of individual membership ($Q$) in R, followed by cluster matching and permutation in CLUMPP (*Jakobsson & Rosenberg, 2007*) to account for label switching artifacts and multimodality in each $K$ tested. We summarized the genetic diversity using a Principal Component Analysis (PCA) in ADEGENET v2.0, which does not make assumptions of HWE and LD (*Jombart, 2008*; *Jombart & Ahmed, 2011*). Finally, we conducted a Mantel test to evaluate isolation-by-distance (IBD) using the standardized genetic distance ($F_{ST}/1 - F_{ST}$) and the logarithm of the geographic distance among sampling sites. To identify significant correlations, Pearson's correlation coefficient, $r$ was calculated in the software ZT (*Bonnet & Van de Peer, 2002*), which it is a program specifically designed for conducting the Mantel test. We used 10,000 permutations to obtain a $p$-value and we plotted the correlation among all locations, and excluding the South Georgia Island, the most differentiated location (see result below). We performed Mantel tests in order to test for two processes that can arise in an IBD pattern: (a) a

continuous cline of genetic differentiation or (b) the existence of well differentiated and disjunct populations (*Jombart & Ahmed, 2011*).

### Recent migration and effective population size

We estimated the percentage of recent immigrants from each of the clusters obtained in GENELAND and STRUCTURE, through an assignment test implemented in the program GENECLASS v.1.0.02 (*Piry et al., 2004*). Immigrants were detected by calculating the likelihood ratio L_home/L_max (*Paetkau et al., 2004*), using a calculation criterion based on allele frequencies described by *Paetkau et al. (1995)*. The probability value was calculated using 1,000 Monte Carlo simulations, using the algorithm described by *Paetkau et al. (2004)* and including an error type I of 0.01.

The effective population size ($N_e$) of each location and number of clusters were determined using the *LD* method (*Waples, 2006*) updated for missing data and following *Peel et al. (2013)*. Values of $N_e$ within corresponding 95% confidence intervals (CI) for each population were estimated using NEESTIMATOR (*Do et al., 2014*) with the following parameters: a minimum allele frequency cutoff of 0.01 and a random mating model. In addition, we estimated a traditional calculation of $N_e$ for a stepwise mutation model (SSM; *Kimura & Ohta, 1978*), following the Nei's formula: $N_e = (1/[1 - H_E]^2 - 1)/8\mu$ (*Nei, 1987*); where $H_E$ corresponds to expected heterozygosity calculated in GENALEX and $\mu$ corresponds to the mutation rates of microsatellites. We used two mutation rates for $\mu$: (a) $1 \times 10^{-2}$ and (b) $1.0 \times 10^{-5}$ mutations / locus / generation, both of which were based on *DeWoody & Avise (2000)*. No mutation rate for microsatellites within the *D. eleginoides* genome have been estimated in the literature; therefore, we chose these broad range of mutation rates reported for marine, freshwater and anadromous fishes in *DeWoody & Avise (2000)* as useful approximations of appropriate rates for *D. eleginoides*.

## RESULTS

### Variability, genetic structure and connectivity

The six loci that we used showed high variability (Table 1, Table S2). The expected heterozygosity for loci ranged from 0.033 (To5) to 0.953 (cmrDe9), and the number of alleles fluctuated between two and 30 (To5 and cmrDe9, respectively) (Table S2). In assessing HWE equilibrium, we found that some sampling locations showed significant deviations in some loci after the Bonferroni correction ($p \le 0.008$): cmrDe4 in SP; cmrDe9 in IQ; cmrDe30 in IQ, FI, and SGI; cmrDe2 in PW; and To5 in FI (Table S2). Pairwise $F_{ST}$ and $R_{ST}$ index showed a significant difference between individuals between the locality SGI and the rest of the sampled locations from South America (Table 2). The values of $F_{ST}$ and $R_{ST}$ index from SGI were one order of magnitude higher than those of the other localities.

### Number of clusters and isolation by distance

The probability distribution provided by the GENELAND program to estimate the parameter "$K$" showed a highest value of $K = 2$ and did not indicate the presence of ghost populations (*Guillot, Mortier & Estoup, 2005*). This indicates that it is highly likely that there are two *D. eleginoides*' genetic clusters or populations in the Southeast Pacific

**Table 2** Pairwise $F_{ST}$ and $R_{ST}$ indices estimated between sampling locations for *D. eleginoides*.

| | NP | SP | IQ | GP | PW | DRI | FI | SGI |
|---|---|---|---|---|---|---|---|---|
| NP | – | −0.02182 | −0.00778 | −0.03261 | −0.03691 | **0.03706** | **0.03472** | **0.12774** |
| SP | 0.00000 | – | −0.00701 | −0.01565 | −0.02354 | **0.04527** | 0.03644 | **0.1356** |
| IQ | 0.00000 | 0.00000 | – | −0.01785 | 0.00348 | 0.01044 | **0.0225** | **0.13586** |
| GP | 0.00006 | 0.00044 | 0.00019 | – | −0.02658 | −0.03441 | −0.02356 | **0.10399** |
| PW | 0.00009 | 0.00011 | 0.00007 | 0.00051 | – | −0.02831 | 0.00566 | **0.09404** |
| DRI | 0.00008 | 0.00006 | 0.00003 | 0.00050 | 0.00017 | – | **0.02089** | **0.09272** |
| FI | 0.00000 | 0.00000 | 0.00012 | 0.00022 | 0.00007 | 0.00006 | – | **0.18169** |
| SGI | **0.00369** | **0.00356** | **0.00367** | **0.00355** | **0.00352** | **0.00355** | **0.00344** | – |

**Notes.**

Here, $F_{ST}$ values are shown below the diagonal and $R_{ST}$ values are shown above the diagonal, with estimates $p$-values of $P < 0.001$ shown in boldface (after Bonferroni correction).

Abbreviations: NP, Northern Peru; SP, Southern Peru; IQ, Iquique; GP, Gulf of Penas; PW, Puerto Williams; DRI, Diego Ramírez Islands; FI, Falkland Islands; SGI, South Georgia Island.

and Southwest Atlantic Ocean. The posterior probability ranged between 0.9–1 (Fig. 2A), supporting the following geographic clusters: The largest cluster, which included the localities of northern Peru, southern Peru, Iquique, Gulf of Penas, Puerto Williams, Diego Ramírez Islands and the Falkland Islands; and the smaller cluster including only the South Georgia Island (Fig. 2A). Likewise, Evanno's index ($\Delta K$), STRUCTURE software, found the same two genetic clusters (Fig. 2B). The Principal Component Analysis (Fig. S1) showed that samples from South America were more similar to each other than samples from the South Georgia Island cluster. The correlation performed to evaluate isolation by distance between geographic and genetic distances was not significant ($rho = 0.089$; $p$-value = 0.603), even when we excluded the comparisons given by the South Georgia cluster ($rho = -0.194$; $p$-value = 0.326) (Fig. S2).

## Recent migration and effective population size

A total of 12 putative migrants were detected in all samples (2.8% of individuals). The number of immigrants from the obtained clusters ranged between 2 and 8 individuals in the smallest (i.e., SGI) and largest cluster (i.e., South America), respectively. Each cluster showed a high percentage of self-assignment, with 89% of the SGI cluster including individuals from the South Georgia Island and 99.3% of the South American cluster composed by locations from South America this clearly supported patterns of genetic structure indicated by GENELAND and STRUCTURE. The same pattern of genetic structure was also supported when analyses were performed based exclusively on sampling locations (Table S3). We detected a predominant migration of individuals from the South Georgia Island to the South American platform (Table S3).

The $N_e$ based on linkage disequilibrium was variable across locations, and ranged from 133 in the Gulf of Penas to infinite for the localities of southern Peru, Diego Ramírez Islands, and the Falkland Islands (Table 1). Confidence intervals for each estimate included infinite values in almost all locations, except around the South Georgia Island. The estimation of $N_e$ based on the clustering analysis showed an infinite value for the South American cluster (Table 1). Conversely, using the formula of Nei while assuming the SMM

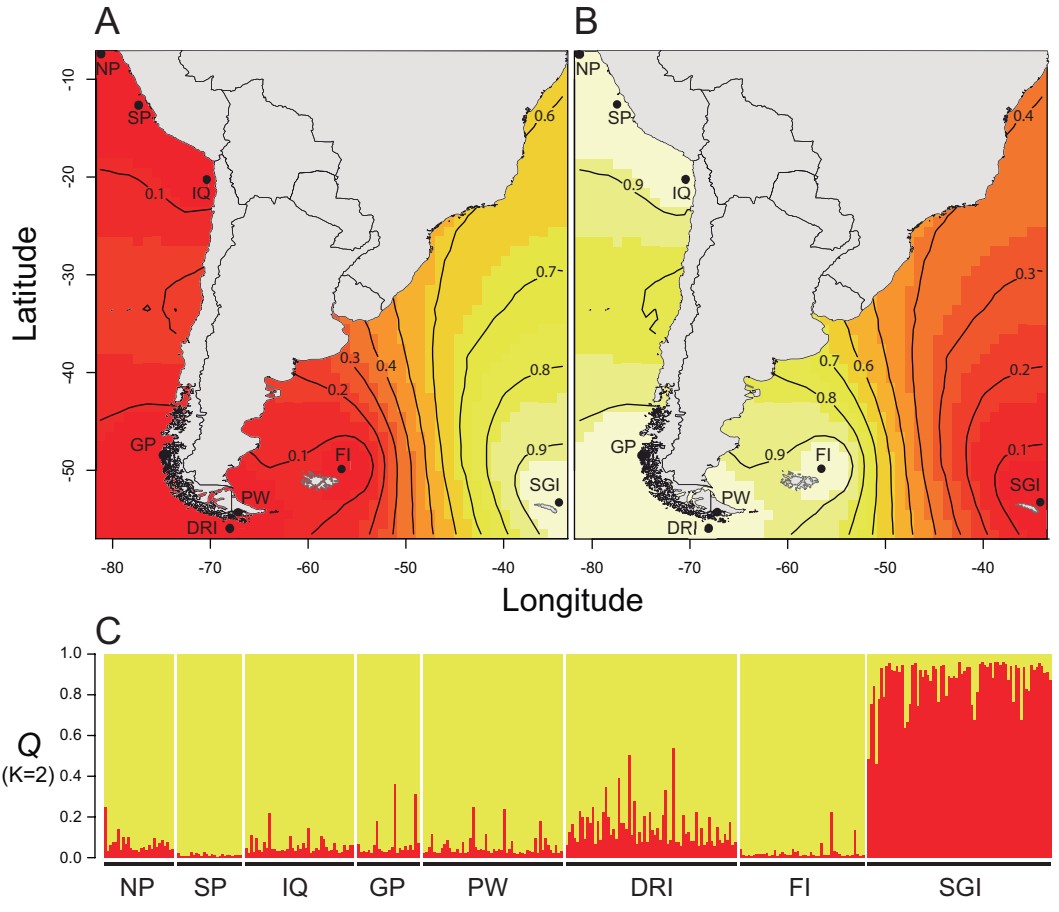

**Figure 2  Results of Bayesian clustering analyses used to infer the number of genetic cluster (*K*) within *Dissostichus eleginoides*.** (A–B) Posterior probability isoclines denoting the extent of genetic landscapes inferred in GENELAND. Clusters indicated by GENELAND included the South American cluster (A), and the South Georgia cluster (B). Black dots represent localities analyzed in this study and regions with the greatest probability of inclusion are indicated by white, whereas diminishing probabilities of inclusion are proportional to the depth of color (increasingly darker red colors). (C) STRUCTURE results showing the estimated population admixture coefficients (*Q*) for each individual, whose genome is broken into colored segments representing the proportion of that individual's genome derived from each of the *K* inferred clusters. Abbreviations: NP, Northern Peru; SP, Southern Peru; IQ, Iquique; GP, Gulf of Penas; PW, Puerto Williams; DRI, Diego Ramírez Islands; FI, Falkland Islands; SGI, South Georgia Island.

(*Kimura & Ohta, 1978*) and either of the mutation rate values discussed above, the northern Peru location had the highest $N_e$ values and the South Georgia cluster showed the lowest $N_e$ (Table 1). The maximum calculated $N_e$ value, for northern Peru, was 2.73 times greater than the minimum calculated value for South Georgia Island.

## DISCUSSION

Overall, our results support a lack of genetic structure among the populations of *Dissostichus eleginoides* inhabiting the South American continental plate, but we infer strong population genetic structure between populations of this area and those of the Southwest Atlantic

Ocean. We concluded that the continuity of the deep-sea habitat along the continental shelf and the biological features of the study species are plausible drivers of intraspecific population genetic structuring across the distribution of *D. eleginoides* on the South American continental shelf.

## Genetic diversity and genetic divergence

Based on six microsatellites loci and an array of complementary analyses, *Dissostichus eleginoides* showed two well differentiated genetic clusters within the study area (Fig. 2), which also showed qualitative differences in genetic diversity parameters (i.e., $H_E$; Table 1, Table S2). The genetic variability of *D. eleginoides* measured by $H_E$ ranged from 0.650 to 0.779, being higher in locations from the South American continental shelf (i.e., $H_E = 0.691$–0.779) compared to the South Georgia Island (i.e., $H_E = 0.650$) (Table 1, Table S2). These values oscillate close to the variability obtained using microsatellites in marine and anadromous fishes (i.e., $H_E = 0.68$–0.79) described by *DeWoody & Avise (2000)*. Previous studies have shown similar values of average $H_E$ by location (i.e., 0.708–0.804 *Appleyard, Williams & Ward, 2004*; 0.800–0.890, *Shaw, Arkhipkin & Al-Khairulla, 2004*; 0.671–0.867, *Rogers et al., 2006*; and 0.788–0.966 *Araneda, 2017*).

Individuals from the SGC showed less allele number and privative alleles than the SAC (Figs. S3A, S3B). These results might be associated with differences in sample size by cluster, however this pattern is also consistent when comparing by location. By contrast, these results suggest that the low genetic variation exhibited by the SGC could be explained by the infrequent movement described for this species between areas (see section below) or by the retention of early stages. The distribution down the shelf slope close to the South Georgia Island and Shag Rocks has been described as a spawning area (*Agnew et al., 1999*) and the Antarctic Circumpolar Current and the Polar Front might isolate early stages of the SGC from locations on the South American Continental Shelf (i.e., SAC in this study). Therefore, an enclosed population could be affected by genetic drift, which changes allele frequencies through time and thereby fixing alleles in this population as seen in our results.

The two well-differentiated clusters are located (i) on the South American continental shelf (i.e., SAC) and around to the South Georgia Island. Along the South American continental shelf, two biogeographic breaks have been described consequence of the currents (e.g., 41°S) and upwelling patterns (e.g., ~30°S and 36°S) of these areas which has been correlated with changes in species composition (see review by *Camus, 2001*; *Fenberg et al., 2015*) and genetic isolation of marine taxa (e.g., *Tellier et al., 2009*; *Brante, Fernández & Viard, 2012*; *Canales-Aguirre et al., 2016*). These patterns do not seem to play a role in the population structure of *D. eleginoides* and its distribution on the South American continental shelf (i.e., SAC). Conversely, our results suggest that the deep-sea habitat continuity on the South American continental shelf, in addition to the inherent biological features of *D. eleginoides* should be key factors explaining the lack of genetic differentiation across this large area. The genetic cluster around the South Georgia Island (i.e., SGC) is clearly isolated from the cluster associated with the South American continental shelf (i.e., SAC). The continuity of the sea floor of these two clusters are separated by abyssal depths (>1,500 m depth; *Shaw, Arkhipkin & Al-Khairulla, 2004*), the Antarctic
Circumpolar Current (ACC), and the Antarctic Polar Front (APF). The SGC coincides with previous studies (*Shaw, Arkhipkin & Al-Khairulla, 2004*; *Rogers et al., 2006*; *Toomey et al., 2016*), reinforcing the hypothesis that the habitat discontinuity in this area acts as barriers to gene flow. Including this result, *D. eleginoides* is genetically structured in to four populations around the world: three of them located in the Southwest Pacific (Macquarie Island), Southern Ocean (South Georgia), and sub-Antarctic islands and seamounts of the Indian sector (*Appleyard, Ward & Williams, 2002*; *Appleyard, Williams & Ward, 2004*; *Shaw, Arkhipkin & Al-Khairulla, 2004*; *Rogers et al., 2006*; *Toomey et al., 2016*), including South Georgia Island (this study); and one large population located on the South American continental shelf. The results obtained in this study fill a gap in knowledge associated with the population genetic structure of *D. eleginoides* distributed across the Southeastern Pacific Ocean.

## Recent migration and effective population size

Connectivity within localities and clusters could be explained by the reproductive characteristics of the species and physical oceanographic features. Early life stages of *Dissostichus eleginoides* are distributed at around 500 m depths and can spend six month in pelagic waters (*Evseenko, Kock & Nevinsky, 1995*; *North, 2002*). These early life pelagic stages have a high dispersal potential, and their transport along the coast of South American could be driven by the Humboldt Current to the north, while the Cape Horn Current to the south (see Fig. 1). This passive dispersal potential could explain the numbers of obtained putative migrants that belong to the different sampling locations (Table S3). We found a low and asymmetrical first-generation migration pattern between the SAC and the SGC; where eight individuals from SAC were found in the SGC and two SGC individuals in the SAC (Table S3). Low number of migrants has also been reported in previous studies based on genetic markers and mark and recapture methods (*Des Clers et al., 1996*; *Appleyard, Ward & Williams, 2002*; *Williams et al., 2002*; *Marlow et al., 2003*; *Shaw, Arkhipkin & Al-Khairulla, 2004*). For example, *Williams et al. (2002)*, using mark-recapture methods around Macquarie Island and Heard and McDonald Islands, showed that 99.5% of individuals were captured at about 15 nautical miles or less from their point of release and only one individual further away (see also *Møller, Nielsen & Fossen, 2003*); demonstrating poor effective migration by adults. The putative migrants that we identified were mainly assigned to close localities from their sampling sites (Table S3), supporting the hypothesis of low dispersion rate suggested by *Williams et al. (2002)* in *D. eleginoides*. Furthermore, *Shaw, Arkhipkin & Al-Khairulla (2004)* noted that along with the Antarctic Polar Front (APF), depth and the large distances that separate these two populations play an important role as connectivity barriers between the South Georgia Island and the sites located around the Falkland Islands. These factors would also limit the dispersion of eggs and larvae, and therefore they would function as the main inhibitors of genetic exchange between populations of *D. eleginoides* from the SAC to the SGC, and vice versa. Nonetheless, the Antarctic Circumpolar Current (ACC) can explain the asymmetrical migration from SAC to the SGC given that it has a clockwise direction. Thus, individuals that go into the
AAC may move from west to east (*Rintoul, Hughes & Olbers, 2001*); however, to test this hypothesis further mark-recapture studies should be conducted.

The $LDN_e$ estimates were not very informative because of the infinity values estimated, which have been suggested to be the consequence of large populations (*Waples & Do, 2010*). Small amounts of LD caused by drift in populations with $N_e$ larger than 1,000 and a low number of genetic markers may explain the estimates in this study. This method assumes random mating, and no immigration, admixture or overlapping of generations (*Waples & Do, 2010*). In our study, we can discard admixture because we found two well differentiated clusters. However, we cannot discard immigration and overlapping of generations. This suggests that the estimation and interpretation of $N_e$ is very challenging when assumptions are violated (*Waples, 1990*). Based on Nei's formula, the $N_e$ estimated for the South Georgia Island showed a lower value than locations on the South American continental Shelf. The $N_e$ for the South Georgia Island was 2.73 times smaller than that for Northern Peru, and 1.89 times smaller than that for the South American cluster (Table 1). These outcomes could be explained by habitat availability (*Venier & Fahrig, 1996*), where there is a continuous continental shelf from Peru extending south round Cape Horn and extending out around the Falkland Islands, whereas the shelf around the South Georgia Island is clearly much smaller. This habitat availability is directly related to the abundance and distribution of the species.

Finally, our results support a large population on the South American continental shelf that is genetically differentiated from the population around the South Georgia Island, and which is potentially the product of the habitat continuity across this area and the inherent biological features of *D. eleginoides*. These results are an important contribution to the further development of management models and conservation plans for this fishery. Moreover, we highlight the need for an international and/or coordinated management strategy for this resource by the different countries involved in the fishery on the South American continental shelf.

## ACKNOWLEDGEMENTS

The authors are very grateful to three anonymous reviewers who greatly improved the final version of the manuscript. The authors are very grateful to the research scientist Matthew Lee for the proofreading English manuscript. We are grateful with the Instituto del Mar del Peru (IMARPE) for providing samples from Peru.

### Funding

This work was funded by Fondo de Investigación Pesquera (FIP 2006-41); and supported by the Fondo Nacional de Desarrollo Científico y Tecnológico de Chile (FONDECYT grant number: 1140692, 1170815, 3150456 and 1170486). Sandra Ferrada-Fuentes was supported by doctoral fellowships for the 'Programa de Doctorado en Sistemática y Biodiversidad', from the graduate school of the Universidad de Concepción, Chile and by a CONICYT

doctoral fellowship. The funders had no role in study design, data collection and analysis, decision to publish, or preparation of the manuscript.

## Grant Disclosures

The following grant information was disclosed by the authors:

Fondo de Investigación Pesquera: FIP 2006-41.

Fondo Nacional de Desarrollo Científico y Tecnológico de Chile: 1140692, 1170815, 3150456, 1170486.

CONICYT doctoral fellowship.

## Competing Interests

The authors declare there are no competing interests.

## Author Contributions

- Cristian B. Canales-Aguirre conceived and designed the experiments, performed the experiments, analyzed the data, wrote the paper, prepared figures and/or tables, reviewed drafts of the paper.
- Sandra Ferrada-Fuentes conceived and designed the experiments, performed the experiments, analyzed the data, reviewed drafts of the paper.
- Ricardo Galleguillos conceived and designed the experiments, contributed reagents/materials/analysis tools, reviewed drafts of the paper.
- Fernanda X. Oyarzun wrote the paper, prepared figures and/or tables, reviewed drafts of the paper.
- Cristián E. Hernández analyzed the data, contributed reagents/materials/analysis tools, wrote the paper, reviewed drafts of the paper.

## Data Availability

The raw data is included in the Supplemental Files.

## Supplemental Information

Supplemental information for this article can be found online at http://dx.doi.org/10.7717/peerj.4173#supplemental-information.

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
