# Peer review of "Population genetic structure of Patagonian toothfish (Dissostichus eleginoides) in the Southeast Pacific and Southwest Atlantic Ocean"

_PeerJ, doi:10.7717/peerj.4173_

## Round 0.1 · original submission · Major Revisions

Dear Authors,

I have now received three reviews. Two recommended major revision and one minor revision. I am recommending major revision.

All three reviewers provided detailed comments and suggestions on how to improve the article, and I would like you to follow these suggestions. When you send in a revision, if you choose to resubmit this MS, please provide a detailed response to the reviewer’s comments as well.

In my own opinion, as presented, the article has a number of major weaknesses, many of them addressed by the reviewers, but here are the principal points I am concerned with.

1) Same as reviewer 1, I question how some of the results were obtain. For example how is it possible that 100% are assignable to a population, yet 2.9% of them are migrants (i.e. belong to another population). There are also issues with interpretations of results. For example how can the authors claim subtle and cryptic genetic differences among populations, but no statistical tests support this assertion.
2) Same as reviewer 2, I agree that the results are over-generalized. You should focus on your species and your study area. There is no need to try to generalize to all deep see fishes in the World’s oceans.
3) The paper needs to be edited for style, grammar and orthography. It should also be shortened and focused to make it easier to read.
Overall, I think the MS and the data have the potential to be an important contribution to understanding of population structuring in an economically important fishery species. However, the MS still needs many improvements, and thus I recommend major revision.

I look forward to seeing a revised version of this MS in the near future.

Sincerely,

Tomas Hrbek

Reviewer 1 ·

Basic reporting

The manuscript spends considerable time discussing geographic and hydrological 'seascape' features that end up having no influence in the data. In particular, these features were not mentioned with an specific hypothesis for how the results would be expected to appear if any particular feature was important, which leads to post-hoc narration, especially considering the authors find no population structure along the South American continent, so these features simply contribute to unsubstantiated theorizing. I think the description of the seascape needs to be condensed.

The English is passable (intelligible) in most cases, but it needs to be edited by a native speaker for grammar, syntax, and spelling. There are a few cases where I was not able to follow what the authors meant, but mostly just errors that distract, some of which I will note below (not comprehensive).

The data have been included as a supplemental file. I do not know if this conforms to journal policy or if they need to be transferred to Dryad.

Experimental design

The authors use six microsatellite loci, which to some degree are affected by null alleles or other genotyping errors, to examine the population structure of Dissostichus eleginoides along the coast of southern South America (including Falkland Islands, FI) and South Georgia Island (SGI). They discover significant population structure between the continent and SGI, and claim to find “subtle and cryptic genetic difference” among locations along the continental coast although I see no evidence of significant structure in their results. The discovery of structure between the continent (including FI) and SGI is not new: it has been shown since Shaw et al. (2004), as the author’s acknowledge, while the examination of structure along the continental coast is novel and of greater interest. I do not believe that the lack of structure precludes these new data from being of general interest, and there is simply no solid evidence that there is structure among the continental and FI sampling localities. While I agree that there is a phantom pattern in the Structure data, 1) it has not been quantified and tested for significance and 2) it seems more likely to be to be a result of using the locprior with the uncorrelated allele frequencies model, which may often introduce artifacts*. In contrast, the IBD test** and pairwise Fst show no significant divergence between the northern and southern localities. Moreover, the largest continental Fst value (albeit with no confidence interval beyond just ‘not significant’) is between GP and PW, while between the most distant localities (NP and FI) the Fst is 0.


I also find it disconcerting that the magnitude of Fst found between SGI and the other localities is almost an order of magnitude lower than found by the other studies to include these localities (Shaw et al. 2004, Rogers et al. 2006). I wonder if this may be a result of null alleles in the data. This also may be affecting the inference of population sizes and past changes in population size. First, the inference of population size from gene diversity is totally different from linkage disequilibrium, except that they are generally larger in larger populations. One (gene diversity) measures long-term population size, while the other (LD) measures contemporary effective number of breeders. As the authors point out, it would not be surprising for the SGI population to have a smaller population considering the smaller available area and habitat compared to the continent. Do these values correlate to catch data? However, I do not see any other trend towards low population sizes from north to south, as the authors claim: in fact, the smallest continental population sizes were the middle localities, so there appears to be no “trend.” Also, considering that there are several instances of null alleles in the data, including at SGI, I would be very wary of interpreting the apparent heterozygote deficiency at SGI as a history of population expansion. Also, I do not even see heterozygote deficit in the data; according to Table 1, observed heterozygosity is larger than expected at 4 of 6 loci!

In addition, regarding null alleles, the authors mention that they examined the data for genotyping errors (e.g. with Microchecker), but it is unclear to me how actually this was done. It is my recollection that Microchecker only checks for null alleles (putative violations of Hardy-Weinberg equilibrium), not genotyping error. This is generally done by repeated genotyping of the same sample followed by comparison of inferred genotypes. If this was done here, please state, and if not, please describe how genotyping errors (and their overall rate) was identified. More broadly, just the sheer variation in the number of individuals genotyped at different loci at each localities is worrisome to me. The authors need to explain why this variation is so great. Further, the authors need to more thoroughly justify why (if) their results are robust to null alleles since no correction was made and the only analysis that supposedly accommodates null alleles in Geneland.

With regards to the inference of connectivity, it is clear based on the Fst values that 1) the connectivity between SGI and the continent should be very low, and 2) that the connectivity among FI and continental localities could/should be non-zero. While the observed values for SGI appear low (2-3%), and ARE low with respect to demographically meaningful values (e.g. for fisheries), they are actually quite high given the magnitude of population divergence (Fst) between these areas. 2% per generation immigration would rapidly erode population divergence. Given that Geneclass is known to give spurious results when the data are weak, I find these values hard to believe. For example, how is it possible that 100% of individuals can be assigned back to the continental cluster but 2.9% are putative migrants? Moreover, first-generation migrants should show up as clearly identifiable in the Structure results, but I see none.

*why was the uncorrelated frequencies model used? These populations with recent common ancestry, no doubt experiencing gene flow, almost certainly have correlated allele frequencies (and admixture). The authors need to justify the use of this model or reanalyze their data with the correlated model.

**Why was this correlation used rather than a Mantel test?

Validity of the findings

see above comments

Additional comments

Please address the following issues:

Figure 1 is interesting, but at the very least needs to have the sampling localities shown. Moreover, I would have preferred to have seen a figure that showed the native range of Dissostichus eleginoides and summarized previous findings as far as genetic population structure is concerned. Considering that so few of the features depicted in Figure 1 are actually implicated in the results (and some of which are unlabeled, e.g. CHC), it seems kind of a waste. Also, “a)” is superfluous if there is no “b)”

Line 83: what is “low fecundity”, what is this relative to? Whales have low fecundity. Most broadcast spawning bony fishes do not. What sort of fecundity limitation are we talking about?

Line 88: taxon is singular, taxa is plural.

Line 129: please correct “can act like barrier promoting genetic divergence as it have been showed in several researchers”

Line 144 DID not instead of DO not

Table 1. I don’t think most of these data (for individual loci) need to be included in the main body of the text. I recommend including coordinates, sample sizes, and multi-locus measures of observed and expected heterozygosity in Table 3 and moving Table 1 to supplemental.

Line 296: Estimating Ne using LD cannot be used to estimate the number of clusters directly, and is only effective indirectly in limited circumstances (e.g. when confidence intervals are fully bounded and not overlapping). This does not apply in with the current results

Line 345 and Figure 3 I don’t understand what the authors mean by this statement (i.e. “subtle similarity of its genetic diversity”), and if I interpret it correctly, I do not agree with that conclusion. Yes, when an additional cluster is added, it largely is found in the northern localities, but also in the Falkland Islands, which is not mentioned. A more appropriate way to examine structure within the continental cluster would be to remove the SGI individuals and re-run Structure (although the Evanno et al. method generally fails when the true K=1). However, given the Fst results, I would surmise that no significant structure will be found.

Line 349 and Figure 4. While what the authors say about PCA is true (that it provides an analysis free of the constraints of HWE and LD), it generally only provides a more easily interpretable rendition of what structure and Fst tell. And the present case confirms this. Thus I think that this analysis and figure are redundant with other analyses, and only serves to clutter the manuscript. I recommend moving it to the supplemental or eliminating it entirely.

Figure 5. Does not show anything of particular note. Move to the supplemental.

Line 405: as I mentioned, there is no test to substantiate this trend, and without the SGI point (which is in a separate cluster and may be spurious anyway), there is no trend to speak of.

Line 407-413: I find this result rather incredible given the observed vs. expected heterozygosity for this locality in Table 1. Moreover, the result may partially derive from modeling with a stepwise mutation model, so I would ask the authors to substantiate how well an SMM model fits these data? Do the alleles form a continuous distribution or is it multi-modal, suggesting that the SMM is inappropriate (and favors a two-phase model).

Figure 6. I don’t find this figure particularly informative beyond a textual description of the results (as has been included, questionable though they are). I recommend moving it to supplemental.

Line 431: I find this statement to be unsubstantiated in the results

Line 432: Even if there were structure (which I do not see), this is post-hoc narration of the data, since to explicit test of any of the mentioned geographical features was made. This would require predicting the form that population structuring would take in response to a given feature, then testing fit to that model (aside from the ACC, which, given previous studies, should have been a foregone conclusion).

Line 438: “-in a conservative way-“ is confusing; I assume that authors mean that ‘…Dissostichus eleginoides showed, conservatively, two well-differentiated…”. Also, general practice is for a genus name to not be abbreviated the first time it is used in each paragraph.

Lines 454-460: see above comments.

Lines 461-483: Post-hoc narration based on an unsubstantiated pattern; needs to be removed, unless the authors wish to describe why they EXPECTED to find a pattern that did not it present itself, at least with these data (there is always the possibility that further data collection will find significant structure).

Lines 492-510: I found this section hard to follow. The authors skip from four populations (which the data seem to support, the continental plus FI population being one of them) to six, including a division of the continental population into two, without any explanation. The continental division is unsubstantiated, but couldn’t follow where the remaining structure comes from. Again, a figure summarizing previous results (including Toomey et al. 2016 Antarctic Science, Volume 28, Issue 5) would be useful.

Line 550: see above comments regarding this “trend”. It would be interesting to compare abundance or catch data from SGI versus continental localities to know if these values are reflected there also

Line 567: I think this is spurious, and the authors make no explanation (a priori or post-hoc) for why an expansion would be expected for these fishes. If anything, given increasing fishing pressure, I would have expected a decline.

Line 573: sentence fragment

Reviewer 2 ·

Basic reporting

# MAJOR COMMENTS

Canales-Aguirre et al. present a study of the population genetic structure of Patagonian toothfish (Dissostichus eleginoides) populations from sub-Antarctic waters over continental shelf habitats around the South American cone eastward to South Georgia Island. The goal of this study was to use genetic markers to infer potential causal mechanisms responsible for population genetic structuring in marine fishes of these sub-Antarctic waters including part of the Southern Ocean, using D. eleginoides as a study system. The authors obtained a large sample of D. eleginoides from a small number of populations across part of the geographical range of the species and evaluated population structure and demographic history using traditional population genetic analyses of microsatellite markers. Overall, the authors generate a molecular dataset that contributes to our knowledge of genetic variation and geographical structure of economically important D. eleginoides populations. But, while the paper could become suitable for publication in PeerJ in the future, significant improvements need to be made first. My recommendation is therefore Reject with encouragement to resubmit after a major revision.

The paper is fairly well written, considering that English is not the authors’ first language. However, a number of changes and improvements to the written English will be required before the paper can become acceptable for publication. These include many minor edits to improve grammar and orthography, as well as more difficult passages that need to be restructured or completely re-written. I have outlined a number of these changes below, but my comments may not be exhaustive and there may be other errors I have not identified. The authors should carefully inspect their paper so that changes are made following these recommendations throughout, where necessary. *Please have a native English speaker edit the manuscript prior to resubmission to avoid any further English issues.* There are other minor issues with notation, tables, and figures that I also address and suggest improvements for below.

However, I find the following three issues more troubling:
#1) the authors demonstrate problems of focus from the Abstract forward and poorly formulate and test their hypothesis;
#2) the authors do not accurately and straightforwardly present existing knowledge of genetic structure in the focal taxon, based on previously published studies, or properly cite other studies in the Methods;
#3) the paper is too long, relative to what I would consider warranted by the content and size of the dataset/sampling (though this can be partly alleviated by fixing #1 above).

## Issue #1: Focus and Hypothesis Statement

Within the first two lines of the Abstract, the authors need to set their focus on the study region/system of interest; as written, these two lines are too vague (though this can be OK for an opening sentence). The Abstract is weakened by not zeroing in on the question and region of interest (continental shelf waters of the southeastern Pacific and southwestern Atlantic around South America, plus part of the Southern Ocean) soon enough and leaving the second sentence too open since. For example, the study area is not the only region with deep-sea habitat. The entire ocean (~>90%) is largely made up of deep-sea zones, with only a minor percentage being comprised of shallow photosynthetic or aphotic zones.

*The section on Lines 72-76 echoes with the same problem of lack of precision seen in the opening lines of the Abstract. I don’t think using the focal species as a model system for addressing such a general question will give results that will be broadly applicable to all deep-sea fishes.* You are simply not going to be able to do as you propose and “generate an overall microevolutionary scenario for how the [deep-sea] species… have originated and changed over time” by looking only in this area of the world’s oceans. In fact, the authors’ conclusions are unlikely to be general to any of the three oceans that they have sampled, because they have only sampled a very small portion of each one. They haven’t even sampled the Southern Ocean except for one sampling locality at South Georgia--the Southern Ocean extends all the way around Antarctica! *As a result, the authors would do well to focus on setting up a hypothesis for their study area, principally around continental shelf zones of South America over to South Georgia, and then test that hypothesis(es), and then only attempting to generalize to that area. The authors should re-write this section and the next paragraph to take up this focus (e.g. using the available literature to propose and test a biogeographical hypothesis for the study area).*

Going further along these lines, *the section at Lines 123-137 seems to contain the hypothesis the authors really want to test, as well as potential mechanisms driving population structure in the focal taxon. However, the hypothesis could be stated more clearly, and this section also suffers from redundancy, as paragraphs above it.* Specifically, sentences on Lines 128-133 are redundant and should be unified to a single, cohesive (non-redundant) whole. These lines, coincidentally, also show **very poor** English. This section has not been carefully proofread, and this and surrounding sentences must be rewritten. Please let a native English speaker review this section after re-writing, prior to next submission.

Looking down in the Discussion, there are two other sections that it might help the authors to move up into the Introduction, in a new section formulating a better hypothesis(es) (and predictions); those sections are found at original Lines 441-454 and Lines 467-478. Moving some of this up to the Intro could help improve the hypothesis statement, and it could also reduce some of the redundancy in the text. Importantly, bringing some of this material up to the Introduction might allow the authors to form a priori hypotheses about expected patterns of population structure/demography in the southeastern Pacific Ocean area of their sampling. This is **very important** as this is the main area of novelty this study brings in terms of sampling and the advance it makes in our understanding of genetic variation in the focal taxon--apparently, no other studies have ever evaluated genetic structure in D. eleginoides in this region using suitable genetic markers.

## Issue #2: Flawed Literature Review and Citing

The literature review presented in the Introduction is flawed in several ways, some related to the poor formulation of the hypothesis. First, I do not recommend that the authors base their study on an essentially unaccessible book or white paper from Oyarzún et al. (2003), as discussed on Lines 98-105 and elsewhere. Perhaps this is the only resource available. However, it is not possible for the reader to quickly or easily verify these claims, because the paper is not available in the primary literature or online. Fortunately, I can offer a solution. In order to avoid the reader simply having to take your word for it, I recommend, especially if the Oyarzun et al. (2003) data belong to the fourth author (who goes by a similar/same name), that the authors reprint a figure or table from Oyarzun et al. (2003) as a supplementary file showing that D. eleginoides in fact shows no population structure in the southeast Pacific Ocean.

Second, the authors present a rather disingenuous and inaccurate accounting of previous genetic studies of the focal species, D. eleginoides, by Shaw et al. (2004) and Rogers et al. (2006). These two studies are only tangentially mentioned, if at all, in the Introduction, are not used to formulate any hypotheses a priori. Much later, however, they resurface during the Discussion, where the authors state they recover some of the same patterns of population structuring as Shaw and Rogers, on Lines 489-491. This trend must be reversed, and a clear and accurate review of these studies must be provided before I will look at the paper again. The authors would do well to formulate a priori hypotheses based on results of Shaw et al. and Rogers et al., and then test those hypotheses using unlinked genetic markers.

In addition to the above issues, the authors have not properly provided citations for the microsatellite loci that they studied. Although they cite one previous study by Appleyard and colleagues on Lines 194-197, this is not the study in which the loci were developed, which are not cited. The authors should cite the following two papers (citations given below) in which the loci were developed. This is open-access publishing, so we have room for more citations (but the paper is still too long, overall, as noted under Issue #3).

Citations to add:

Reilly A, Ward RD (1999) Microsatellite loci to determine stock
structure of the Patagonian toothfish Dissostichus eleginoides.
Mol Ecol 8(10):1753–1754

Smith P, McVeagh M (2000) Allozyme and microsatellite DNA
markers of toothfish population structure in the Southern
Ocean. J Fish Biol 57(Suppl A):72–83


## Issue #3: Paper Is Too Long

The paper is far too long, at >30 pages and probably >10,000 words, given the number of loci and only partial sampling of the species geographic range, in addition to partly negative results. There are some sections where the authors cut cut the most, especially the Discussion, which is about 1-2 pages too long. The first paragraph of the Discussion section is a brief outline/overview of what is discussed below it, without strongly stating support or rejection of the hypothesis. SO, the authors could simply delete this paragraph. However, in that case, I think the description of congruence between genetic differentiation and province and ecoregion boundaries should be kept, only moved further down in the Discussion.

The authors can also cut length by cleaning up the writing so it is not so redundant, and by focusing in on one or two very specific hypotheses in light of the biogeographical context of the study area (only).

Experimental design

Given other studies of D. eleginoides have been published based on combined analyses of 5-7 microsatellite loci and mtDNA gene sequences (e.g. 12S) (Shaw et al. 2004; Rogers et al. 2006, cited in main text), it is somewhat puzzling that the authors do not follow suit here and include data from mtDNA sequences as part of their study. Moreover, given that >10-12 years have elapsed since these previous studies, I was suprised to find that the authors are still using a smaller number of loci is used than I find to be common in most such studies, which often employ at least 10 microsatellite markers. I was also very surprised that mtDNA genes were not included, even from a partial sampling, given that fish mtDNA genomes are readily and cheaply sequenced today.

These are not damning issues, but I wonder why the authors have designed their study this way, and I would like to hear a response or discussion on that, for example why 6 microsatellites is rigorous enough to address the questions that they are interested in.

Two aspects of the experimental design that partly overcome the limited number of loci are 1) that the loci are informative and 2) that the authors sampled many individuals within each population, apparently from 24-71 individuals. So, this should give them good sampling properties for inferring allele frequencies etc. The methods, as written, also seem quite replicable, and I am pleased to see that the data are included in an appropriately formatted supplementary file (avoiding their placement in online repositories, although this might also be a good idea; for example, the authors could store the data in Dryad or Zenodo).

Nevertheless, the authors do a poor job of discussing the need for mtDNA or other marker types to be integrated with their data to improve their inferences. In particular, I felt the statements on Lines 482 to 483 are right but could be expanded upon. For example, what do these other data types offer that microsats do not?

Validity of the findings

The novelty of this paper is partially undercut by the fact that previous studies that used more spatially and numerically (in terms of nucleotide / locus) extensive sampling also arrived at similar conclusions (e.g. Rogers et al. 2006). However, this paper contains what appear to be statistically valid analyses, and thus is consistent with Aims & Scope of the journal. See other sections where I have made recommendations about the hypothesis-testing approach used in the paper, and how to improve on it.

Additional comments

# MINOR COMMENTS BY SECTION

## ABSTRACT

See Major Comments above. After these are fixed, I will thoroughly review English in the Abstract; however, here are a couple suggestions:

Line 27 – Causal mechanisms “form” or “are responsible for” or “underlie” phenomena observed in nature, they are not “of” those phenomena. Change “of” on this line to something like “potentially underyling…”

Lines 28-29 – Your study area isn’t really these regions, but rather is confined to continental shelf waters around the southern tip of South America and only one site within the Southern Ocean proper. Change the wording here so that it doesn’t sound like your study area is larger than it really is.

Line 35 – Change “Georgia” to “South Georgia” in order to make your writing more specific. Georgia is a state in the USA and a small country in the Caucasus of Eurasia; don't leave any possibility of being confused with these, not even for a millisecond.

## INTRODUCTION

Line 48 – This should say, “…that populations of deep-sea animals with broad geographic ranges had low genetic variability.” That’s what you really mean. Otherwise it might read as though you meant deep-sea animals should also have broad ranges.

Line 49 – Change for improved readability. This should say “it has often been assumed” or “workers have often assumed…” This line would also be improved by saying “speciation in the deep sea”; there is a worrying pattern of breaking sentences with commas. You are not speaking; you are writing a scientific paper, so it’s better not to have so many of these.

Line 50 – I think you mean “geographic isolation by distance”, rather than “geographic distance”. Please change this.

Line 51 – Change “that marine organisms have” to read, “of marine organisms.”

Line 53 – You really want to contrast the preceding sentences with the information that is given starting on this line. So, why not start this sentence with “However, during the past…”? The authors also need a comma after “decade” here.

Line 54 – Don’t use so many little words. Delete “the” wherever it is not necessary. We should strive for pithy, concise scientific writing, and doing this will help you achieve this goal. Delete “the” before the word “increasing” on this line.

Line 55 to 56 – Change “In addition” to “For example,”. Besides that, this is too wordy. Delete “other”, change “the organisms” to “organisms” (delete little words that don’t matter!), and simplify the end by changing it to read, “may restrict population connectivity”. There is also a level of imprecision here—what “physical changes” are you referring to? Change “physical changes” to “environmental change” or something along those lines.

Line 59 – If a species has population structure, then that is obviously a phenomenon that occurs within its range! Delete “in the geographic range of the species.”

Line 61 – Delete the comma after (Ne) and change “Ne” so that the “N” is in italics (it is an estimated parameter) and the “e” is lower case. Do this *throughout*.

Line 63 – Here, we find a sentence that is completely out of place. This should be the third-from-last sentence in the preceding paragraph. Move this sentence up to the end of Line 55, just after the Etter et al. (2005) reference. Also, I would recommend shortening it to focus only on how these gradients influence genetic structure, since the first part of it (“At present, deep-sea environments are regarded as highly complex ecosystems”) is redundant with Lines 54 to 55. Continuing on the theme of redundant sections that need to be revisited, I realized at Line 67 to 68 that the authors were re-writing the same thing over and over, that bathymetric gradients promote population structure. This is highly redundant, so this section needs to be cleaned up to avoid being redundant.

Line 67 to 68 – No comma is needed after the closing parenthesis (delete that comma).

Line 68 – Make this clearer. A better antonym for sessile is “vagile.” I would suggest changing “some studies of mobile vertebrates” to read, “some studies of vagile marine vertebrates”. Note that you should change “on” to “of”. Also, change “fish” to “fishes” or “fish species,” because “fish” is often used to indicate a single species (the other terms are more inclusive, referencing multiple fish species, and also sound better in my opinion).

Line 71 – change “and potentially increase” to “, potentially increasing” (note I added a comma and you should too).

Line 77 – Add a comma after “study”.

Line 79 – Don’t call it a “commercial resource”, it is a fishery. Also, please don’t undersell this species. The Patagonian toothfish fishery is the most productive and lucrative (i.e. economically important) fishery in the entire Antarctic, Southern Ocean, and southern portions of the oceans around the southern South American cone. As far as I am aware, there is no fish or invertebrate that is more popular in the region or as an export to Europe and North America.

Line 81 – Also due to its life history, including relatively small numbers of eggs and delayed onset of reproductive maturity—I know, you say the latter in the following sentence but you omit the small clutches (add them please).

Line 85 – Instead of “found”, I think you mean that this species is typically fished at these depths. The Patagonian toothfish spends most of its life cycle in less than 300 m of water, then larger juveniles and adults move into deeper water to reproduce. It’s wrong to say individuals of the species are typically found in the deep sea. It’s mainly in the epipelagic seas, in terms of total time throughout the lifetime of an individual.

Line 86 – Be consistent with hyphens. Use “deep-sea environment” throughout (as you have been doing), or just “deep sea environment”—use one, but not the other.

Line 92 – Avoid vagueness, which can confuse the reader. You just referred to two species in the genus. So, you can’t just use “it” here. Change “it” to read “D. eleginoides”.

Lines 92 to 98 – This is a long and *confusing* sentence!! Readers may struggle to find the second, complementary statement relative to “While”, because the sentence is too long. Please fix this by breaking this into two sentences.

Line 93 – Again, delete unnecessary little words. Delete “its”.

Line 101 – Change “challenges” to “challenge.”

Line 104 – Be specific. Start this sentence by referring to isozymes, not “These molecular markers.”

Lines 106 to 107 – You’ll want to focus on what really matters here—genetic, demographic, and geographic processes, not simply “environment and the biology of species.”

Line 124 – Change “be” to “present”.

Line 134 – What does “South American distribution” mean? It’s not distributed in South America, but along the coast/continental shelf waters there. This appears to be a prediction of the authors’ hypothesis and it is poorly stated.

## MATERIALS AND METHODS

Line 144 – “do not” and “obtain” are present tense; however, M&M should almost always (with few exceptions--see below) be written using past tense. Please fix this.

Line 154 – You will want to add that this species is not listed under CITES.

Lines 157 to 158 – Sorry, not true! You did not sample “the entire distributional range” of the species. You sampled the portion of the species range around South America and South Georgia, but you conveniently fail to mention that the species has an extensive distribution across the southern Atlantic Ocean and Indian Ocean, which you didn’t sample.

...

Lines 302 to 303 - This is one of the exceptions to the past tense rule for M&M sections I mentioned above: here, you need "corresponds" instead of "corresponded", because the symbol/abbreviations were, are, and will remain representations of specific parameters.

Lines 304 to 305 - You need to change µ to "mutation rates". Next, delete the next instance of mutation rates and replace with "of which." So, I think the sentence, as rewritten, should look like this:

"We used two mutation rates for μ: a) 2.5 x 10-4 and b) 1.0 x 10-5
mutations / locus / generation, both of which were based on Bos
et al. (2008)."

The real problem with this however is that you do not provide sufficient explanation of why you chose these rates. Please provide additional detail, including a justification of why you believe these rates are valid for the loci and taxon you studied.

## RESULTS

...

## DISCUSSION

Line 425 - Remove little words that clutter your writing. You could improve this sentence by deleting the "a" before "strong." You also need to fix the same error at other points in the text, including Line 468.

Lines 427 to 428 - Keep it simple. You could change "analyzed geographic area" here to simply "study area."

Lines 429 and 430 - Need to refer to authors with year in parentheses here, in both cases... as in, "Camus (2000)... Spalding et al. (2007)."

Line 431 to 432 - Change "subtle" to "subtly" and do not refer to "locations," because you are referring to the populations not the areas in Chile.

Line 434 - Look at the present-tense word "generate" here. This should be past tense ("generated"), but also remember that there is a distinction between processes that have generated geographic population structure, and processes that maintain that structure.

Line 438 to 439 - Change the dashes/hyphens for commas. This format is unusual and very, very sparingly used in academic writing, and usually avoided instead. Also change "differentiate" to "differentiated" to correct verb tense.

Lines 439 to 440 – There are number and tense problems here. Change “genetics” to “genetic”, and also change “described” to “describe”. Check verb-tense agreement in this section.

Line 454 - Why is Camus 2001 cited here? Probably should remove this citation.

Line 457 – “STRUCTURE software does not revealed(space)” looks like an unfinished line. Please complete this sentence.

Line 459 – Things are not separated “for” a distance, they are separated “by” a distance. Change this to read something like, “separated by over 2,900 km from the nearest sampling locality…”

Line 461 - You don't mean the entire southeast Pacific Ocean, you mean the portion of that ocean in your study area; change the text to reflect this.

Lines 462-464 - It is my opinion that the end of the sentence that starts the paragraph needs to be fixed to say that the upwelling pattern has been correlated with changes in species composition and genetic isolation of marine taxa, so as not to imply that you know that it caused these, to have proof of mechanism (we work in a probabilistic framework). Also, remove the comma after the closing parenthesis on Line 463.

Lines 466 and 471 – You do not need to hyphenate cold water, unless it is written to modify the word that follows the phrase (e.g. cold-water fusion). Thus, remove the hyphen between “cold” and “water”.

Line 467 – Delete “their” before “populations.”

Line 468 - Delete the comma after "inodorus," which is unnecessary.

Lines 472-474 – It looks as though translating from Spanish to English did not work well here. As a result, the order of the clauses in the sentence is jumbled. Please rewrite this sentence to improve clarity by placing "as a barrier promoting genetic isolation" after the word "suggested."

Line 483 - Need to define abbreviations at first point of use. Here, since you don't really use this abbreviation much in the paper (and not after this point of use), you should simply write out the meaning of SNP, rather than using the abbreviation. Change "like SNP" to ", such as single nucleotide polymorphism data, "...

Line 492 - Simplify and clarify the writing by changing, "of D. eleginoides, its," to read, ", D. eleginoides."

Line 514 - The populations don't "have" connectivity per se; instead, the populations appear to be connected, or exhibit evidence of connectivity, based on your results.

Lines 529 to 530 - Again, avoid using these hyphens to separate clauses. I suggest that you place the clause between hyphens inside of parentheses instead.

## TABLES AND FIGURES

Tables are pretty good, as constructed; however, I'm not sure they adequately note where p-values have been given Bonferroni's correction. I think this information should be included/referred to in the table captions.

The figures are beautiful and mostly well designed and appropriately labelled. However, K is an estimated parameter, not a known quantity, so it should be italicized in mathematical notation. Thus, go through the figures and text and make sure K is in italics throughout (see similar comments elsewhere (e.g. Table 1 comments below) for similar recommendations for Ne and other popgen summary statistics/parameters). This was a striking mistake, since "Q" was appropriately placed in italics in the figures and captions (e.g. caption to Fig. 3), while "K" was not.

Figure 1 - The Cape Horn Current (CHC) is not labelled on the diagram. Please add the abbreviation in the appropriate place. This figure could also be improved by adding country names for South American countries shown, as well as the name for nearby islands.

Table 1 Locality column – It is not appropriate to only given latitude and longitude coordinates to the level of detail of degrees followed by minutes. This is insufficient. Coordinates should be given as degrees, minutes, seconds, or in decimal degrees format. Please change this.

Table 1 Parameter column – Here, and throughout the entire manuscript, for NA, Ho, and He, these parameters should be placed in italics because they are estimated parameters rather than known values. For Ne, for effective population size, it’s the same thing: place the “N” in italics, as I mentioned above.

Table 2 caption - Bold values do not represent p-values at all. Fix this by stating that the Fst values with p-values of P < 0.001 are shown in boldface.

Table 3 - "MIG_PUT" is not an appropriate abbreviation for this variable. Use just "Mig." or "PPM" or something logical, not a variable name with an underscore. Also at line 392 within the caption, I think you meant to refer to populations instead of localities, and also you should state that these localities are on the South American coast or continental shelf.

Figure 5 - The "F" in "Fst" and the "p" in "p-value" must always be italicized because these are estimated parameters. Change this throughout the manuscript (e.g. also in the text on Lines 352, 353, 407 and 408, and others).

Figure 6 - The dots/symbols for data points in this figure are small and hard to see or differentiate. Please make the symbols more legible by making them larger and filled in. Also, following along the same lines as above, the "H"s in "He" and "Heq" should be given in italics in the figure and caption.

Reviewer 3 ·

Basic reporting

#The text is in general very clear but need some minor adjustments as provided to the authors.
#The authors showed a large biological background on the species studied and also a good background on abiotic features afecting the phenomenon studied.
#Figure 1 could be improved. My suggestion is to provide the sampling sites on in. All other Figures are ok.
#The raw data (msats genotyping) was not provided.
#One table could be moved to Suplementary material as mentioned to authors at General comments for the author.
#The defense of the hypothesis is the critical point in the text. But I am sure the authors can offer a better and focused view on the phenomenon detected. My suggestions are also provided at the General comments for the author.

Experimental design

#The research is very original and is fitted to the journal.
#The questions are clearly posed.
#The investigation was well performed.
#Methods were tiresome describe. The authors need to provided a table with the loci genotyping among individuals.

Validity of the findings

#The findings are very interesting an important for fishery management.
#The data are very robust.
#Clear conclusions and hypothesis are the critical points in the text. But I am sure the authors can offer a better and focused view on the phenomenon detected. My suggestions are also provided at the General comments for the author.

Additional comments

Sugestions:

INTRODUCTION
LINES50 -53: It is important to comment briefly that currently there are lots of evidences that marine environment is not so homogeneous in terms of genetic discontinuities. It is necessary to include around 3 papers from 2014, 2015 and 2016 on this issue.
LINE63: "At present????…(Chase et al. 1998; Etter et al. 2005; Zardus et al. 2006)." The youngest citation with 11 years. It needs to be modified.
LINES63-76: The authors need to update the references used. For example, why not to use AMY R. BACO, RON J . ETTER, PEDRO A. RIBEIRO, SOPHIE VON DER HEYDEN, PETER BEERLI and BRIAN P. KINLAN (2016). A synthesis of genetic connectivity in deep-sea fauna and implications for marine reserve design. Mol.Ecol 25, 3276–3298????
The antiquity of the papers is repeated throughout the text and it must be altered.
LINES 133-134: The phrase "Therefore we expected find that D. eleginoides populations are genetically structured in its South American distribution" must be removed. The main hypothesis of the study is very clear and the authors do not need to improve it with a sub-hypothesis as posed.

M&M
LINES152-154: The phrase "No specific approval of this….." is unnecessary and might be deleted.
LINE208: Please correct "…only one loci in the location GP…" to "…only one locus in the location GP…".
LINE210: Please correct "Although (Chapuis & Estoup 2007)) proposed…" to Although Chapuis & Estoup (2007) proposed…"
LINE215: Please substitute "sibship" to "kinship".
LINES215-216: Please substitute "full-sibs" to "total kinship" in the whole text.
LINE219: Please substitute "outcomes" to "results"
LINES236-240: It is important to measure the RST parameter because it indicates specific differences for msat loci.
LINES275-277: The phrase "This kind of multivariate method…" is unnecessary.

Important: The authors must use the Rst parameter. This is specially addressed to a msat dataset. The Rst values can be obtained easily in Arlequim. The authors can also improve the table 2 offering the Rst values up of the diagonal.

In general the text can be optimized by excluding lots of method explanations. Most of the text offered by authors in M&M refers to method explanations. In general the community knows the general basis of those methods used.

RESULTS
LINES410-413: Results must have only the description of the obtained results. The phrase comprising these lines indicates a conclusion. Explanations, conclusions, and new hypothesis are expected to be in the Discussion section.

Table 3 must be better edited. It was almost impossible to understand it.
Figure 6. I suggest this figure to be moved to supplementary materials.

DISCUSSION
Line 457. Please change from "cryptic genetic" to "incipient genetic".
Lines459-460. Change the phrase "…and are separated for more than 2,900 km of distance from the next sampled locality (i.e. Quellon)." by "…and are separated for more than 2,900 km from one region (Northern and Southern Peru) to another (Iquique)."
Line 477. After Wares et al. 2001, delete the word "these".
Lines 478-481. The authors are a little confused to explain the genetic structure observed between the northern and southern pacific samples of Patagonian toothfish. In one moment they argument in favor to a IBD (isolation by distance…2900km in distance). However they provide the IBD test that failed to support it. In another moment of the discussion the upwelling zone would be the support for the genetic division within those demes. Based on the evidences provided by authors I suggest they argument in favor to incipient allopatry by upwelling zone.
Line 540, 541. change "infinity" by infinite"

The topic "Contemporaneous demographic history" is very confuse and must be revised. Previously authors showed high migration rates and in the line 546 is written "In our study we can discard immigration given the scarce number of estimated migrants,…". I suggest the authors to be focused. At several points in the discussion the authors insist to explain methods. This is not adequate.
Another feature of this text is the double-hand stile. The authors start explaining something in a way of thinking and immediately after that they change the way of thinking offering another explanation. It is also inadequate. The authors need to defend what they actually believe. The evidences are very clear and I do not understand why the text is sometimes so confuse.
Lines 557-563 This veer long phrase "Consequently, the South Georgia cluster has a higher probability of being affected by stochastic processes (e.g. environmental changes, demographic or genetic changes) and therefore, has a higher risk of extinction (Palstra & Ruzzante 2008) and loss of genetic variation (Frankham et al. 2003). Stochastic factors are commonly initiated by deterministic factors (e.g. natural selection or
harvesting; Palstra & Ruzzante 2008), like fishing efforts that could affect and reduce the population size of D. eleginoides in its Southeast Pacific and Atlantic Ocean distribution." is not correct.
I did not understand the relationship between habitat availability (at this case a large area) provided by authors. Species with small habitat size in general are more susceptible to those mentioned stochastic events. If not, why conservation biologist as us are so concerned o understand the genetic structured populations? Closed or limited gene flow in terms of habitat availability has more impact than in those species occurring in large habitat availability as showed by authors.
Lines 567-569: "Finally, we suggest that D. eleginoides has recently experienced a population expansion in the Georgia cluster, based on the significant heterozygote deficiency obtained in BOTTLENECK"… is this conclusion correct? Based on another methods regarding tests on historical demography it is expected that populations experiencing expansions has more variation than contrary. Please revise it.

---

## Round 0.2 · Major Revisions

Dear Authors,

I have now received two reviews. One accept and one major revision. I also recommend major revision.
I think the MS has been improved significantly in terms of analyses and interpretation of results, but there still remain many problems in how the MS is structure, i.e. orthography, grammar, style. I completely under reviewer two’s frustration. Many of the issues have more to do with carelessness than with anything else. How can the MS contain three different misspellings of the focal species family of this MS? The genus name is also not always spelled correctly. Why are species names not capitalized? There are Spanish abbreviations, etc. This shows lack of attention when preparing the MS, or the attitude that the reviewers and the editor will fix this.
So please implement the suggestions of reviewer 2, and when you return the MS, make sure that any addition errors and oversights not pointed out by the reviewer were also fixed.

Sincerely,

Tomas Hrbek

Reviewer 2 ·

Basic reporting

I consider this paper to be a marginal "pass" in the category of "Basic reporting," but this "pass" score is conditional on the fact (and _required future outcome_) that basic presentation needs to be improved before the paper could become suitable for publication. I give a more detailed list of suggestions for helping with this below, under the "General comments for the author" section; here I explain my score for Basic reporting.

Under the category of Basic reporting, I appreciate that the literature references and background information provided in the revised manuscript are much better than that of the initial submission at the beginning of the year. Also, given that the authors have followed my suggestions to re-write the paper to have a clearer focus on the study organism/area and to clearly state an a priori hypothesis of no genetic structuring within the focal taxon, I would generally agree that the paper is now mostly self-contained, with results that bear directly on the hypothesis.

However, the English has a number of orthographic errors, and exhibits unprofessional aspects of content in writing and the table and figure captions. Second, and more importantly, the paper as written does not adequately describe or discuss the results, with problematic sections mainly being the Abstract and Discussion, but also part of the Introduction.

Experimental design

No comment.

Validity of the findings

No comment.

Additional comments

# MAJOR COMMENTS

Canales-Aguirre et al. present a study of the population genetic structure of Patagonian toothfish (Dissostichus eleginoides) populations from sub-Antarctic waters over continental shelf habitats around the South American cone eastward to South Georgia Island. I reviewed the first submission of this manuscript, in which the goal of the study was to use genetic markers to infer potential causal mechanisms responsible for population genetic structuring in marine fishes of these sub-Antarctic waters including part of the Southern Ocean, using D. eleginoides as a study system. This goal was overstated and over-generalized, and I suggested that the paper needed to be re-written due to this and other issues including a number of glaring problems with the written English. After pointing out these shortcomings, I summarized three major concerns in my previous review, as follows (paraphrasing): *Issue #1:* problem of focus and poor formulation and testing of the hypothesis(es) of the study; *Issue #2:* inaccurate and inadequate review of existing knowledge of genetic structure in the focal taxon based on previous studies available in the literature; and *Issue #3:* paper length not warranted by content and size of the dataset/sampling and scope of study (paper too long; related to #1 as well).

Regarding Issue #1, I commend the authors for re-writing the paper to align with some of the main aspects of my suggested focus—i.e. focusing on the focal species alone, and the study area within which that focal species was sampled. I have also stated this in my assessment of the paper under the Basic Reporting category of the review form. Moreover, the authors have improved the paper by explicitly stating the hypothesis in the Abstract and Introduction.

In response to Issue #2 that I raised previously, the authors have re-written parts of the paper, especially in the Introduction, to provide a more suitable account of the state of knowledge of genetic variation and population structure within D. eleginoides at the onset of their study, and I appreciate that aspect of the revised manuscript. Also, as a result of my initial comments and those of Reviewer #1, the authors have deleted a number of problematic sections (containing potentially spurious analyses or weak conclusions/weakly supported patterns) from the manuscript, and to an extent this has alleviated issues revolving around my point raised under Issue #3 above. The length of the paper is now suitable for publication.

Nevertheless, I still find several issues that keep the revised manuscript from being suitable for publication at this time, and, in particular, issues arise because (1) I find the authors’ handling of Issue #1 has been less than satisfactory—they need to better relate their results (e.g. in Abstract and Discussion sections) back to the hypothesis, and (2) there are a number of issues with the written English and presentation that preclude the paper from being suitable for publication. Quite frankly, I am slightly angered that the authors would state multiple times in their rebuttal letter that “Matthew Lee native speaker and Marine Biology Ph.D.” has proofread the new version of the manuscript, so that “this new version should not have any typography error. Other issues associated with notation, tables, and figures were also checked.” This is just *_ridiculous_*. As noted below, I found _two errors with the English in the first line of the Abstract alone_. If Matthew Lee has given the final revised manuscript draft an in-depth read through and actually provided corrections, then only one of three possible scenarios could be true: (1) the English in the revised draft is poor because he did a _terrible job_ (I can assure you, his performance was really awful in this case, and you should *never use him again for proofreading*); (2) Matthew Lee did a good job, but you didn’t follow his recommendations (shame on you); or (3) Matthew Lee did an OK or good job, but you chose to go on your own and edit the paper substantially afterwards, and many errors crept into the manuscript during this period. The next time that someone recommends you to take your manuscript and have it fixed by a native English speaker, please do a better job and present a better final draft than what you have provided in this revision! And it’s probably best that you _not_ use Matthew Lee!!

Due to these issues, I consider this manuscript to warrant a decision of Reject with encouragement to resubmit a paper with Major Revisions. The only reason I feel that this will be a Major Revision, and not a Minor one, is that I agree with the journal description of this type of Recommendation, in that I would “prefer to re-evaluate any revised version,” and my suggested edits for correcting the English (below) are extensive.

# MINOR COMMENTS

As a general suggestion, never submit a Word document to an English journal that has the language setting set to Spanish. This indicates to the Editor and Reviewers that your experience during writing was one in which the Spanish spell checker, and not the English spell checker, was used during writing. Moreover, this has obviously caused you to miss several mistakes that a spell checker in English could have provided you with. So, when writing, and when revising papers, *always* perform the following procedure when creating the document:
(1) select all text/content in your .doc or .docx file (i.e. press ctrl + A);
(2) click on Tools > Languages…;
(3) select English (US) if the paper is destined for an American journal (one that uses American English conventions) and select English (UK) if the paper is destined for a British journal (or one that uses British English spellings); and
(4) then save the file and go back through it looking at possible errors highlighted with underlining in the text (possible misspellings or grammatical errors).
**This is very easy to do, and everyone should take the responsibility to do it.**

Another **general issue** is that the authors have managed to screw up the italics and subscripting throughout the manuscript. The big letters in parameter names like He, Ho, Ne, etc. are always italicized. Fix throughout. For He and Ho, it is probably more common that the subscripts be in uppercase; so make that change too. Also, these are not just "parameters", they are "summary statistics" or "statistics of DNA polymorphism."


## ABSTRACT

Abstract Line 1 – How can the authors, in their rebuttal letter, possibly claim that a native English speaker has proofread the manuscript, when the _first line_ of the Abstract contains two errors?? The errors are as follows:
- You need to change “population genetics” to “population genetic”
- Also, the name of the focal species is misspelled as “Disosstichus eleginoides”. Please fix to read “Dissostichus eleginoides”!

Abstract Lines 3 & 4 – Problems here with tense and the word “distribution” that must be fixed. The word “focus” should be replaced with “have focused” and the authors shouldn’t be discussing distribution of D. eleginoides as the focus of previous studies—they should be stating that these were genetics studies. I strongly recommend the following rewording, “…studies have focused on the genetics of D. eleginoides in the Southern Ocean… knowledge of their genetic variation along the South American continental shelf.”

Other Abstract changes:

- Change “structure on the South American” to read “structure along the South American.”
- Make next sentence read, “We hypothesized that this species would show zero or very limited genetic structure…”
- The very next sentences should read as follows:

“We used Bayesian and traditional analyses to evaluate population genetic structure, and we estimated the number of putative migrants and effective population size. Consistent with our predictions, our results showed no significant genetic structuring among populations of the South American continental shelf, but supported two significant and well-defined genetic clusters of D. eleginoides between regions (South American continental shelf and South Georgia clusters).”

- Change “…these clusters was 11.3%” to read “…these two clusters was 11.3%”.
- Finally, add “D. eleginoides” before “populations along the South American continental shelf” in the last sentence of the Abstract.

## INTRODUCTION

Line 2 – Change “which” to “that”… as in “that remain stable.”

Line 5 – change “isolation by distance” to “isolation-by-distance” with the words separated by hyphens, and also provide the abbreviation for this term here in parentheses. This should read “isolation-by-distance (IBD; Wilson & Hessler 1987).”

2nd Paragraph – I have helped you by re-writing the middle section of the 2nd paragraph of the Introduction to have several minor fixes that it would be complicated for me to write out and explain. Please make the section starting with “In broadly distributed benthopelagic fishes…”, and continuing several lines, read as follows:

“In broadly distributed benthopelagic fishes, considerable gene flow has been reported among populations. Scarce genetic divergence is therefore mainly the result of the availability and continuity of their habitats (e.g. continental slopes, the slopes of oceanic islands, slopes on seamounts), facilitating gene flow (Smith & Gaffney 2005; Jones et al. 2008; Lévy-Hartmann et al. 2011; Varela et al. 2012). In addition, biological features such as highly vagile and/or pelagic adults and long-duration planktonic egg, larvae, and/or juvenile stages are associated with low intraspecific genetic differentiation (Shaw et al. 2004; Rogers et al. 2006).”

3rd Paragraph – First, the correct family name for rockcods of Antarctica is “Nototheniidae” and NOT “Nothothenidae” or "Notothenidae"; please fix this spelling here and throughout (fix elsewhere as necessary). When referring to Nototheniidae at the middle of this paragraph, this would read better as, “…belongs to the Nototheniidae family, a diverse clade of Antarctic and sub-Antarctic origin.”

3rd Paragraph – Change “This continuous distribution along…” to read “The continuous distribution of this species along…”. Also you should probably say that the continuity or connectivity of the populations could facilitate gene flow homogenizing their population structure. I suggest that the first part of the last sentence of this paragraph be changed to read, "The continuous distribution of this species along the South American continental shelf in the Southeastern Pacific Ocean could facilitate gene flow homogenizing their population genetic structure..."

4th Paragraph – First line: population studies are conducted on or of species, not "in" them. Change this line to read, "Population genetics studies of D. eleginoides to date..."

4th Paragraph – Second sentence: before citing "Smith & McVeagh", move the mention of data type to the start of the sentence. Change sentence to read, "Using allozyme and microsatellite loci, Smith & McVeagh (2000) showed that D. eleginoides ..."

4th and 5th Paragraphs - I have too many edits to suggest for this section. In fact, I have more edits than I care to write out line by line. Here I provide a highly edited version of the last two paragraphs that I suggest the authors simply copy and paste into their manuscript:

"Population genetics studies of D. eleginoides to date have been mainly conducted in the Southern Ocean. Using allozyme and microsatellite loci, Smith & McVeagh (2000) showed that D. eleginoides has restricted gene flow between the Falkland Islands, and zones south of the Antarctic Polar Front (i.e. Heard Island, Ross Dependency, Prince Edward Island and Macquaire Island). Later, Shaw et al. (2004) showed that populations to the north of the Antarctic Polar Front (i.e. Patagonian Shelf, North Scotia Ridge) and to the South of the Antarctic Polar Front (i.e. Shag Rocks, South Georgia) have stronger genetic differentiation in the mtDNA genome than the nuclear genome, based on microsatellites and mtDNA sequences. In a study conducted in the West Indian Ocean sector of the Southern Ocean, Appleyard et al. (2004) investigated mtDNA and microsatellite loci but found no evidence for among-population genetic differences associated with islands. Subsequently, Rogers et al. (2006), surveying samples from islands in the Atlantic, Pacific, and Indian Oceans, found genetic differences based on microsatellite and mtDNA data. Specifically, Rogers et al. (2006) indicated that toothfish populations from around the Falkland Islands were genetically distinct to those from around the South Georgia Islands. Recently, Toomey et al. (2016) studied DNA from otoliths and found differences between populations around the Macquarie Islands and other locations surveyed in the Southern Ocean.

All previous studies discussed above have focused mainly on islands of the Southern Ocean, leaving a distinct gap in our knowledge of the genetic structure of D. eleginoides populations across their Southeastern Pacific Ocean distribution. The only study carried out in the Southeastern Pacific Ocean was developed by Oyarzún et al. (2003) based on allozymes and was restricted to a small geographic area. Oyarzún et al. (2003) did not find population genetic structure among samples collected in south-central Chile (c. 37ºS to 43°S). Sampling across a wider geographical area of this region while using more sensitive molecular tools that have higher levels of detection of DNA polymorphism, such as microsatellite loci, could aid in determining whether or not significant population genetic structure exists among D. eleginoides populations across their Southeastern Pacific Ocean distribution.
In this study, we used a panel of six microsatellites previously developed for D. eleginoides to test whether this species shows population genetic structure on the South American Plateau. We hypothesized that D. eleginoides will not show genetic structure due to the continuity of suitable habitats along the South American continental shelf, from Peru in the Pacific Ocean southward and eastward to the Falkland Islands in the Atlantic Ocean (Fig. 1).
"

** Please compare with original text and consider taking all of my suggested edits here. **

## MATERIALS AND METHODS

Ethics Statement – This paragraph would read better if parts of it were changed around, reordered to improve the readability and flow of the writing. In particular, it would be a good idea to mention the point about CITES, and how the sampling strategy obviated the need for collecting permits/government approval, near the beginning of the paragraph. Here is a re-write for you with the suggested change and some fixes to the English:

“Dissostichus eleginoides has not yet been assessed for the IUCN Red List and is not listed under CITES. Samples used in this study were collected in accordance with legislation of the corresponding nations. In fact, no governmental approval of this vertebrate work was required since the Patagonian toothfish individuals sampled in this study were obtained from scientific and commercial fishing activities. We did not kill fishes for the purpose of this study; instead, we obtained tissue samples from individuals that were fished by authorized commercial vessels using long lines. Tissue samples of Patagonian toothfish used in this study were obtained from the Peruvian exclusive economic zone (EEZ) in collaboration with the Instituto del Mar del Perú (IMARPE). Tissue samples from the Chilean EEZ were obtained during scientific research programs with the permission of the Chilean Fishery Government and obtained by the Instituto de Fomento Pesquero (IFOP). Additional tissue samples from the…”

Molecular and pre-processing genetics dataset - Please call the ABI 3730 an "automated sequencer". Also, you should say that you "estimated" or "measured" allele size, not that it was obtained.

One fix for a problematic sentence: "Ultimately, we obtained a total data set of 357 individuals that we used in subsequent analyses."

Further down... instead of saying to "identify total kinship," you should say "To estimate total kinship..."

Another fixed sentence: "Results from the total kinship identification analysis did not show evidence for putative total kinship in the data set; therefore, we proceeded with data analysis without excluding any individuals. "

Genetic variability and structuration index - I don't like the authors' use of "structuration" in the title here. But another issue is the vague use of the term "parameters" here. Please refer to number of alleles and heterozygosities etc. as population "summary statistics". Further down in this paragraph, Markov chain is misspelled... this should be "Markov chain" (chain is lowercase) and not "Marcov Chain". I also fixed another sentence in this section: "No pair of loci in our data set exhibited significant LD, which indicated that all the loci used in this study were independent from one another (unlinked). "

Another issue in this and all sections: FST and RST should always have "ST" shown as subscript text. Fix throughout.

Number of clusters and isolation-by-distance - Here, I have too many edits to write out. Please re-write as follows:

"To infer the most likely number of genetic clusters (K) present in our data set, we used two Bayesian clustering methods, one in the program GENELAND v1.0.7 (Guillot et al. 2005a; b, 2008) and the other implemented in STRUCTURE v2.3.4 (Pritchard et al. 2000; Falush et al. 2003). GENELAND uses a Bayesian statistical population algorithm to model a set of georeferenced individuals with genetic data, while accounting for the presence of null alleles in the sample. The number of clusters was determined by 10 independent Markov Chain Monte Carlo (MCMC) searches, which allowed us to estimate K using the following parameters: K from 1 to 8 (which is equivalent to the number of sampling locations surveyed in this study), 5 x 106 MCMC iterations, a thinning interval of 1,000, the maximum rate of process Poisson fixed at 357, and the maximum number of nuclei in the Poisson-Voronoi tessellation fixed at 1071. Following recommendations of Guillot (2008), we ran the analyses using the uncorrelated frequency allele model, the spatial model, and the null allele model because the value of K in the study area was unknown. Finally, we plotted a map of South America over the output of GENELAND, in order to visualize the results in the context of geography.

Although STRUCTURE does not account for null alleles and uses a non-spatial model based on a clustering method, it is useful for quantifying the proportion of each individual genome from each inferred population in K. The number of clusters was determined by performing 10 runs in STRUCTURE with 50,000 iterations, followed by a burn-in period of 5,000 iterations, for K = 1–9. All STRUCTURE runs were carried out with an admixture model of ancestry, an independent allele frequency model, and a LOCPRIOR model (Hubisz et al. 2009). We incorporated Evanno’s index ΔK (Evanno et al. 2005) in order to identify the best K value for our data set, using STRUCTURE HARVESTER (Earl & VonHoldt 2012). Then, we plotted ‘consensus’ coefficients of individual membership (Q) in R, followed by cluster matching and permutation in CLUMPP (Jakobsson & Rosenberg 2007) to account for label switching artifacts and multimodality in each K tested. We summarized the genetic diversity using a Principal Components Analysis (PCA) in ADEGENET v2.0, which does not make assumptions of HWE or LD (Jombart 2008; Jombart & Ahmed 2011). Finally, we conducted a Mantel test to evaluate isolation-by-distance (IBD) using the standardized genetic distance (FST / 1 - FST) and the logarithm of the geographic distance among sampling sites. To identify significant correlations, Pearson’s correlation coefficient, r, was calculated in the software program ZT (Bonnet & Van de Peer 2002), which it is a program specifically designed for conducting the Mantel test. We used 10,000 permutations to obtain a p-value and we plotted the correlation among all locations, excluding the South Georgia Islands, the most differentiated location (see Result below). We performed Mantel tests in order to test for two processes that can arise in an IBD pattern: a) a continuous cline of genetic differentiation or b) the existence of well-differentiated and disjunct populations (Jombart & Ahmed 2011).
"

Recent migration and effective population size - Looking at the references to methods by Paetkau et al. (2004) and Paetkau et al. (1995), these are _really old methods_. _Are these methods still valid, or have they been replaced by more recent methods that improve on them or fix errors?_ If so, or if newer methods are available for doing this, then why did the authors use older methods? The authors should provide a brief answer and explanation here.

Also in this section, the authors use the term "confidence intervals" for the first time. Later in the paper (e.g. in table or fig captions), the authors abbreviate this as "IC"; however, this is the Spanish abbreviation and is **incorrect** for an English journal. _No one will intuitively know what this means, or shouldn't be expected to do so_. The authors should change this abbreviation to "CI", and it should be given here and used throughout. I suggest changing the sentence mentioning NEESTIMATOR to read as follows, "Values of Ne within corresponding 95% confidence intervals (CI) for each population were estimated using NEESTIMATOR (Do et al. 2014) ..."

Another fixed sentence: "No mutation rates for microsatellites within the D. eleginoides genome have been estimated in the literature; therefore, we chose these broad ranges of mutation rates reported for marine, freshwater and anadromous fishes in DeWoody & Avise (2000) as useful approximations of appropriate rates for D. eleginoides."

Recent migration and effective population size -

Fixed sentences in this section: ". Each cluster showed a high percentage of self-assignment, with 89% of the SGI cluster including individuals from the South Georgia Islands and 99.3% of the South American cluster composed by locations from South America; this clearly supported patterns of genetic structure indicated by GENELAND and STRUCTURE. The same pattern of genetic structure was also supported when analyses were performed based exclusively on sampling locations (Table S3)."

ALSO

"Conversely, using the formula of Nei while assuming the stepwise mutation model (SMM; Ohta and Kimura 1973) and either of the mutation rate values discussed above, the northern Peru location had the highest Ne values and the South Georgia cluster showed the lowest Ne (Table 1). The maximum calculated Ne value, for northern Peru, was 2.73 times greater than the minimum calculated value for South Georgia Islands."

In the above fixed sentence, I suggest that the authors write out the full name of the abbreviated model, referred to as "SMM", using its full name, "stepwise mutation model." Here, I believe the correct citation is Ohta and Kimora 1973. But someone please check this before adding.


## DISCUSSION

The first paragraph needs numerous fixes. Here they are:
"Overall, our results support a lack of genetic structure among the populations of Dissostichus eleginoides inhabiting the South American continental plate, but we infer strong population genetic structure between populations of this area and those of the Southwest Atlantic Ocean. We conclude that the continuity of deep-sea habitat along the continental shelf and the biological features of the study species are plausible drivers of intraspecific population genetic structuring across the distribution of D. eleginoides on the South American continental shelf." Compare this text to the original text to see that I have made several changes here.

Genetic diversity and genetic divergence - Here, I would highlight how an array of "complementary analyses" were conducted on the 6 microsatellite loci, not just "different" analyses.


## TABLES AND FIGURES & THEIR CAPTIONS

Table 1 - Caption could be improved by changing title to, "Mean summary statistics for genetic variability, percentage of putative migrants, and effective population size by location and cluster inferred for Dissostichus eleginoides." Also caption has errors. Try:

"Locality abbreviations: NP, Northern Peru; SP, Southern Peru; IQ, Iquique; GP, Gulf of Penas; PW, Puerto Williams; DRI, Diego Ramírez Islands; FI, Falkland Islands; SGI, South Georgia Islands. Locality code SAC refers to the cluster including all locations that are on the South American continental shelf. The SGI cluster included individuals from the South Georgia Islands. Other abbreviations: CI, confidence interval; Inf, infinite; Lat, Latitude; Long, Longitude; N, Number of individuals sampled; NA, average of the number of alleles per locus, HO, average of the observed heterozygosity; HE= average of the expected heterozygosity; NA, not applicable. Effective population size (Ne) was based on Linkage Disequilibrium (LD) (Waples & Do 2010) and Nei (1987) formula. The migrants (M) column shows the percentage of putative migrants from the first generation. (
* Estimated using a mutation rate of 1 x 10-2 (refs. in DeWoody & Avise 2000)
** Estimated using a mutation rate of 1 x 10-4 (refs. in DeWoody & Avise 2000).
"

Table 2 - Don't use the word "index" (singular). Change the title to, "Table 2. Pairwise FST and RST indices estimated between sampling locations for D. eleginoides." Also, I suggest that the caption be re-written as follows: "Here, Fst values are shown below the diagonal and Rst values are shown above the diagonal, with estimates corresponding to p-values of p < 0.001 shown in boldface (after Bonferroni correction). Abbreviations: NP, Northern Peru; SP, Southern Peru; IQ, Iquique; GP, Gulf of Penas; PW, Puerto Williams; DRI, Diego Ramírez Islands; FI, Falkland Islands; SGI, South Georgia Island."



Fig. 1 – Nice figure. I like the distribution polygon. However, please make several changes. Consider changing the first line to mention this is a map of sampling localities that has all of this other detail. For example, you could change this line to start as, “Map of sampling locations used in the present study…” (highly recommended).

Another issue with Fig. 1 is that the distribution polygon is not clearly explained. Please change the next-to-last sentence to read something like, “The geographical distribution of D. eleginoides on the South American continental shelf was obtained from Aramayo (2016) and is shown in transparent gray shading.”

Fig. 2 - The caption to this figure _must be completely re-written!_ I fixed it for you. Again, there are way too many errors or inconsistencies between fig/tables/text to write out all of the suggested changes here. Here is the fixed version:

"Figure 2. Results of Bayesian clustering analyses used to infer the number of genetic clusters (K) within Dissostichus eleginoides. A) posterior probability isoclines denoting the extent of genetic landscapes inferred in GENELAND. Clusters indicated by GENELAND included the South American cluster (left panel), and the South Georgia cluster (right panel). Black dots represent localities analyzed in this study and regions with the greatest probability of inclusion are indicated by white, whereas diminishing probabilities of inclusion are proportional to the depth of color (increasingly darker red colors). B) STRUCTURE results showing the estimated population admixture coefficients (Q) for each individual, in each cluster. Each vertical bar in plot B represents a single individual, whose genome is broken into colored segments representing the proportion of that individual’s genome derived from each of the K inferred clusters. Abbreviations: NP, Northern Peru; SP, Southern Peru; IQ, Iquique; GP, Gulf of Penas; PW, Puerto Williams; DRI, Diego Ramírez Islands; FI, Falkland Islands; SGI, South Georgia Islands."

Reviewer 3 ·

Basic reporting

The new version was improved and attended the suggestions offered.

Experimental design

The data analyses was improved and attended the suggestions offered.

Validity of the findings

No coments.

Additional comments

The contribution was reallly improved.
The phenomenon tested was now better interpreted given the new analyses provided.
The paper will contribute for a better understanding on the phenomena explainning the historical population history of marine organins living around South America.

---

## Round 0.3 · Minor Revisions

Dear Authors,

I have now received a reviewer who previously recommended a major revision. Based on the reviewer’s and my evaluation of the current version, I also recommend minor revision.

There are still minor issues to be fixes in how the MS is written. The reviewer has pointed most of them out, and I found a couple more (I will send an annotated MS separately). Otherwise the MS is much improved in comparison to the first two versions, the results and conclusions are robust, and overall the writing is much, much better.

If you implement the recommended changes, then I do not see why I will not be able to accept the MS after you resubmit it.

Sincerely,

Tomas Hrbek

Reviewer 2 ·

Basic reporting

Despite many improvements, I find a number of minor inadequacies in the written English and so I provide some suggestions for correcting those under "General comments for the author" below. However, I am confident that after correction the paper could become suitable for publication.

Experimental design

No comment.

Validity of the findings

No comment.

Additional comments

# MAJOR COMMENTS

Canales-Aguirre et al. present a study of the population genetic structure of Patagonian toothfish (Dissostichus eleginoides) populations from sub-Antarctic waters over continental shelf habitats around the South American cone eastward to South Georgia Island. I reviewed the first and second (revised) submissions of the manuscript, and have been pleased to see that this next and 2nd revision is very close to publication quality. However, I feel that the manuscript is a minor revision, as a number of minor issues with the English and other aspects of presentation preclude it from being publishable in its present form.

# RECOMMENDATIONS

Line 28 - Change "their" to "in this species."

Lines 31 to 32 - Change "if" to "whether", and then also change "genetic structure on the South American continental shelf" to "genetic structuring in this region." There is too much redundant mention of "South American continental shelf" in the Abstract.

Line 32 - Change "structure" to "structuring" for added variety and easier reading.

Line 41 - Change "America continental shelf compared with the" to "American continental shelf as compared with". Don't call South Georgia Island "the South Georgia Island", unless island is plural.

Lines 43 AND 691 (check throughout) - Your Spanish speaking roots show through here! Change "intraespecific" to "intraspecific".

Line 109 - several minor issues here. Please re-word text in parentheses to read, "(e.g. slopes of continents, oceanic islands, and seamounts)." One issue was a missing period after the "g" in "e.g."

Line 115 - change "also" to "as well as".

Line 116 - Change "like" to "such as." Some will disagree that this is not an error or inappropriate; however, I would intensely disagree. The over-use of the word "like" in daily English conversation is a common issue with millennials, and many if not most authors and editors of scientific papers have switched to avoid this word so as to keep with technical style. Search a major paper in a high quality peer-reviewed journal such as Molecular Ecology or Systematic Biology (e.g. Near et al. 2011 Syst. Biol. 60(5):565–595) and you will find that the word "like" is usually absent. Instead, it is more common to use "unlikely" or "likely" or "likelihood." And the phrase "such as" is more commonly used in place of "like."

Line 136 - move the "(Bialek, 2003)" reference to the end of this sentence (next line) where it belongs.

Line 142 - Please say "family Nototheniidae" and not "the Nototheniidae family."

Lines 155 AND 177 - Please change "Macquaire" to "Macquarie" in both places here. I think the authors meant to refer to the World Heritage Site island, Macquarie Island, between New Zealand and Antarctica.

Line 169 - Please change "nuclear genome" to "the nuclear genome."

Line 171 - Change "among-populations" to the singular form, "among-population."

Line 172 - Need to add a comma after "(2006)."

Line 189 - Change "will not show" to "would show limited."

Line 233 - No need for page breaks between sections. Please delete this break.

Line 245 - Delete the comma after "Additional," so that this reads "Additional tissue samples."

Line 251 - Please stop adding comma splices to your sentences. Delete the comma after "eleginoides" on this line.

Lines 310 to 311 - Capitalization errors. Several expanded abbreviations are capitalized but their definitions need to be given in lowercase. Fix this by changing "Not" to "not", "Confidence Interval" to "confidence interval", and "Infinite" to "infinite".

Lines 315 and 316 - Both lines should read, "a mutation rate of..." Also, the double parentheses, created by putting "2000" in parentheses, are inappropriate. Simply delete the parentheses immediately around the year 2000.

Lines 422 and 425 - The authors failed to completely fix their misspelling of "Markov chain", which I believe that I pointed out in my review of the first revision of the manuscript. I know you've had to do a lot of edits, but please be a little more careful here. It is correct to put "chain" in lowercase; however, Markov must also be spelled correctly. Thanks.

Line 443 - Add "the" before "Bonferroni."

Line 447 - Change "cluster number" to read simply "clusters" here.

Line 448 - Here, you need to correct the sentence for number agreement. Since you are referring to two methods, then "methods" must be plural. Change "clustering method" to "clustering methods."

Line 452 - Here and several other places in the manuscript you have double period marks, as a result of your editing the manuscript. Please correct this here and throughout by searching and replacing all double periods ("..") with a single period mark (".").

Line 461 - Change "account for" to "include a."

Line 510 - Change "Mantel test" to "Mantel tests," making it plural.

Line 512 - Misspelling. Change "disjunt" to "disjunct."

Line 571 - Change "FST" to "FST values."

Line 607 - Need to make the word "posterior" uppercase here because it is the first word in the sentence.

Line 661 - No need to define the abbreviation for SSM again here; it is already defined at Lines 542 to 543 above.

Line 744 - Change "are" to "is."

Line 773 - Change "well differentiated" to "well-differentiated," with the two words separated by a hyphen. Also change "i.e" to "i.e.", with a period after the "i" and the "e".

Line 782 - Change "the knowledge" to simply "knowledge."

Line 788 - Change "spent" (past tense) to "spend."

Line 793 - Change "migration" to "migration pattern."

Line 801 - Please state this as "...migrants that we identified..."

Lines 802 to 803 - Please state the species name rather than saying, "...in this species."

Line 828 - Change "visceversa" to "vice versa."

Line 868 - Change "which" to "who."

That's all. Good job.

---

## Round 0.4 · accepted · Accept

Dear authors,

I am now happy to recommend acceptance of this MS in its current form.

Congratulations on a job well done.

Tomas Hrbek